# QPoint: End-to-End Lightweight Point Cloud Processing via Robust Quaternion Feature Learning

Zhiming Zhou [1]   Yong He [1]   Chaoxu Mu [1 2]   Qiaoyun Wu [3]   Ajmal Mian [4]

## Abstract

The inherent sparsity, lack of structure, and rotation sensitivity of point clouds often lead to high computational and parameter cost in robust feature learning. To address these problems, we present QPoint, a lightweight framework that leverages robust quaternion feature learning. QPoint incorporates a Quaternion-Enhanced local perception module that uses learnable rotations to stabilize local features against geometric transformations, and a Quaternion global attention mechanism that employs quaternion similarity to capture global geometric context with inherent rotation invariance. Extensive experiments show that QPoint achieves top performance across multiple tasks. It achieves excellent 95.0%, 93.9%, and 92.1% overall accuracy (OA) on the challenging ScanObjectNN variants (OBJ_BG, OBJ_ONLY, PB_T50_RS), 94.7% OA on ModelNet40, and 87.0% instance mIoU on ShapeNetParts. Furthermore, QPoint exhibits superior generalization in few-shot learning scenarios. Crucially, it accomplishes this with extremely minimal parameter and computational requirements, establishing a strong and efficient baseline for point cloud processing. *Our code is available.*

## 1. Introduction

Point clouds provide a direct geometric representation of objects and environments in 3D space, playing an indispensable role in autonomous driving, robot navigation, and augmented reality (Ha et al., 2024; Zhou et al., 2025; Nguyen et al., 2025; Zhou et al., 2026). The rapid progress in deep learning has spurred the development of diverse point cloud processing methods, which can be broadly categorized into

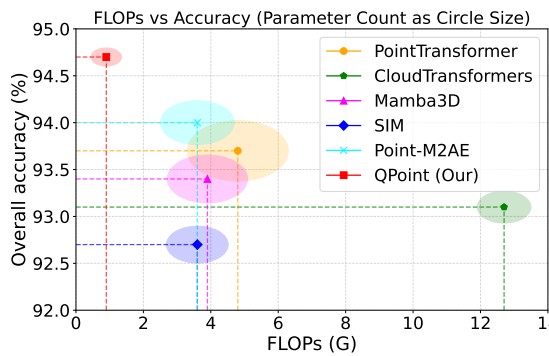

*Figure 1.* Accuracy-FLOPs-Parameter trade-off on ModelNet40. Our QPoint demonstrates the best overall performance in terms of overall accuracy, FLOPs, and Parameter.

projection-based CNNs (e.g., PointCNN (Li et al., 2018)), point-based MLPs (e.g., PointNet++ (Qi et al., 2017b)), Transformers (e.g., PT (Zhao et al., 2021)), and recent Mamba architectures (e.g., PCM (Zhang et al., 2024)). However, the inherent sparsity, unstructured nature, and acute sensitivity to geometric transformations (particularly rotation) pose fundamental challenges. Existing approaches often compromise between performance and efficiency, relying either on extensive data augmentation to mitigate rotational variance or on large model capacities to learn invariance, both of which incur high computational and parameter costs. This limitation severely hinders their deployment in resource-constrained scenarios. Consequently, developing a point cloud processing model that simultaneously achieves rotational robustness, high accuracy, and a lightweight design has emerged as a critical and urgent research objective.

Quaternions (Hamilton, 1844) offer distinct advantages for 3D vision due to their elegant representation of rotations. As hypercomplex numbers, a quaternion $q = [w, x, y, z]$ compactly encodes a 3D rotation with only 4-parameters, providing an efficient and numerically stable alternative to the 9-parameter rotation matrix, while inherently avoiding the gimbal lock problem. These properties have established quaternions as a cornerstone in computer graphics and inertial navigation. Recently, quaternion-based neural networks (Gaudet & Maida, 2018) have emerged, demonstrating the potential to embed geometric priors into deep learning. However, within point cloud processing, their application remains narrowly focused on specific tasks. A systematic framework that deeply integrates quaternion rota-

[1]Anhui University, China [2]Tianjin University, China [3]Nanjing University of Aeronautics and Astronautics, China [4]The University of Western Australia, Australia. Correspondence to: Yong He <h.yong@hnu.edu.cn>.

*Proceedings of the 43rd International Conference on Machine Learning*, Seoul, South Korea. PMLR 306, 2026. Copyright 2026 by the author(s).

tional invariance into feature learning is still lacking, leaving their strong geometric representation potential largely untapped for robust point cloud understanding.

The above discussion positions quaternions as a promising yet underutilized tool. This naturally leads to a fundamental research question: *How can a unified architecture coherently unify explicit local rotational robustness and implicit global rotational invariance through quaternion algebra?* To this end, we propose **QPoint**, a lightweight framework that coordinates learnable quaternion rotations to explicitly stabilize local features against transformations, with a novel attention mechanism using quaternion similarity to implicitly encode invariance into global feature interaction, thereby achieving comprehensive geometric awareness and robustness at minimal computational cost.

Specifically, we design a Quaternion-Enhanced Local Perception Module (QELP) to process local neighborhood point sets. QELP leverages a learnable unit quaternion to adaptively rotate the normalized local coordinates, followed by a residual connection that fuses the rotation-enhanced coordinates with the original coordinates. Remarkably, this process introduces only 4 parameters while significantly enhancing the model's robustness to local geometric rotations transformations. We further propose the Quaternion Global Attention Module (QGA), which projects global features into the quaternion space, normalizes them into unit form, and computes the attention score matrix using the quaternion dot product. This design enables the attention mechanism to directly measure feature similarity in terms of rotational semantics, thereby guiding the model to focus on geometrically relevant information. As a result, it effectively achieves rotation-aware global context modeling. As shown in Figure 1, QPoint attains superior performance in point cloud processing while maintaining extremely low FLOPs. In summary, we make the following four key contributions:

1. We propose the Quaternion-Enhanced Local Perception module (QELP), which improves the rotational robustness of local features at a minimal cost of only 4 learnable parameters, effectively addressing the sensitivity to rotation in prior methods.

2. We design a Quaternion Global Attention (QGA), a novel mechanism that integrates quaternion rotational invariance into self-attention, effectively capturing rotationally invariant long-range geometric dependencies.

3. We propose QPoint, an end-to-end lightweight framework centered on a unified QPoint Block integrating our QELP and QGA modules, delivering holistic rotation robustness for 3D point cloud processing.

4. Extensive experiments validate that QPoint achieves state-of-the-art performance across classification, segmentation, and few-shot learning tasks while maintaining exceptionally low parameters and FLOPs.

## 2. Related Work

**MLP-based Approaches.** Methods based on MLPs constitute a foundational branch of deep learning on point clouds. As a pioneering work, PointNet (Qi et al., 2017a) is the first to process raw point set directly using shared-weight MLPs, addressing the inherent permutation invariance through a symmetric function (e.g. maxpooling). Its successor, PointNet++ (Qi et al., 2017b), introduces hierarchical sampling and groping to capture multi-scale local features. Subsequent research further advances this line of work. PointMLP (Ma et al., 2022) enhances high performance through residual MLP blocks and feature recalibration, while PointNeXt (Qian et al., 2022) demonstrates the continued potential of simple MLP architectures through model scaling and refined training strategies. These methods lay the groundwork for direct point cloud processing, but their capabilities in fine-grained local feature aggregation and global context modeling remain relatively limited.

**Transformer-based and Mamba-based Approaches.** The Transformer architecture (Vaswani et al., 2017) has been effectively applied to point cloud processing, leveraging its strong capability to model long-range dependencies. PTv1 (Zhao et al., 2021) pioneered direct point cloud processing with attention mechanism, PTv2 (Wu et al., 2022) and PTv3 (Wu et al., 2024) have further updated the performance metrics for Transformer-based point cloud processing. But they suffer from quadratic computational complexity in self-attention and large parameter counts, which remain critical bottlenecks for practical applications. Recently, Mamba (Gu & Dao, 2023), a state space model based architecture, has demonstrated strong potential across various domains owing to its linear computational complexity and effective long-sequence modeling. Related pioneering works such as PointMamba (Liang et al., 2024), PCM (Zhang et al., 2025) explore the use of Mamba as a backbone network for point cloud feature extraction. However, Mamba is inherently more suited for processing sequences with inherent order or causal relationships, and its primary only focus remains on computational efficiency.

**Pre-training and Multimodal Embedding Approaches.** Recent advances in point cloud representation learning have witnessed two prominent trends: unified multimodal embedding and generative pre-training. These methods prioritize transferability across scenarios or generative capability, relying on large-scale pre-training paradigms and complex network architectures to learn features. For instance, OmniVec2 (Srivastava & Sharma, 2025) integrates cross-modal sharing to unify 3D and 2D representations, while Point-GPT (Chen et al., 2024) adopts auto-regressive generative modeling for pre-training, while Point-MAE (Pang et al., 2022) adopt masked point modeling for self-supervised pre-training. Point-M2AE (Zhang et al., 2022) further intro-

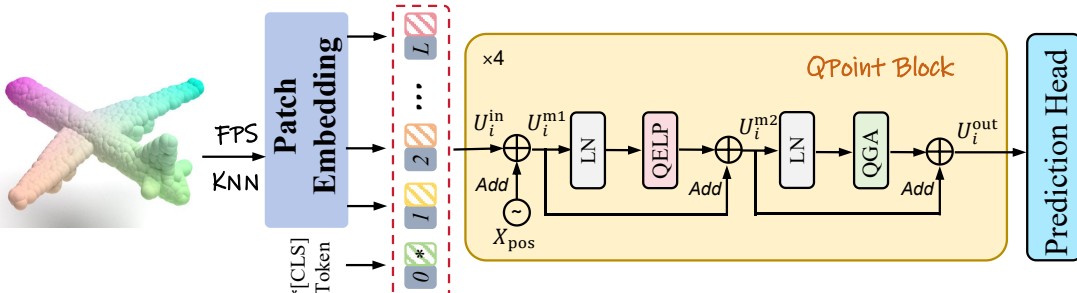

*Figure 2.* The overview of the QPoint framework. The input point cloud is tokenized via FPS, KNN and patch embedding. The tokens are processed by QPoint Blocks (QPB) that model local geometry (QELP) and global context (QGA) for task-specific predictions.

duces a multi-scale masked autoencoding strategy to learn discriminative hierarchical features. In contrast, QPoint is designed as a lightweight end-to-end framework tailored for resource-constrained scenarios. It abandons heavy pre-training pipelines and instead embeds quaternion geometric priors directly into local and global feature learning, enabling rotation-robust inference without relying on external data or multi-modal inputs.

**Quaternion Feature Learning.** Quaternions (Hamilton, 1844) provide a compact and singularity-free representation for 3D rotation. In deep learning, quaternions have been employed to process signals with inherent 3D spatial relationships, with the core idea of embedding geometric priors directly into network structures. Existing studies have explored quaternion-based approaches for tasks such as 3D rotation-equivariant network design (Parcollet et al., 2018; Zhao et al., 2020; Assaad et al., 2023), point cloud registration (Wisdom et al., 2016), and pose estimation (Wolter & Yao, 2018), demonstrating their potential to enhance geometric representation and rotational robustness. However, current works mainly incorporate quaternions into specific network layers, without systematically exploring how their geometric nature can holistically address the interconnected challenges of point cloud disorder, unstructured data format, and rotation sensitivity. Our work aims to fill this gap by fully leveraging quaternions to construct an end-to-end lightweight point cloud processing framework.

## 3. Method

### 3.1. Quaternion Preliminaries

Quaternions constitute a hypercomplex number system that provides a compact and computationally efficient representation for 3D rotations and orientations in three-dimensional space. A quaternion $\mathbf{q} \in \mathbb{H}$, where $\mathbb{H}$ denotes the set of all quaternions, is expressed as:

$$\mathbf{q} = w + x\mathbf{i} + y\mathbf{j} + z\mathbf{k}, \quad \mathbf{i}^2 = \mathbf{j}^2 = \mathbf{k}^2 = \mathbf{ijk} = -1, \quad (1)$$

where $w \in \mathbb{R}$ is real coefficients part and $x, y, z \in \mathbb{R}$ are three imaginary parts, and $\mathbf{i}, \mathbf{j}, \mathbf{k}$ are the fundamental quater-

nion units obeying the given relations. The quaternion is often represented in vector form as $\mathbf{q} = [w, \mathbf{v}]$, where the scalar part $w$ represents the rotation angle, and the vector part $\mathbf{v} = [x, y, z]^\top$ defines the axis of rotation. Thus, A unit quaternion ( $\|\mathbf{q}\|_2 = \sqrt{w^2 + x^2 + y^2 + z^2} = 1$ ) corresponds to a valid 3D rotation.

A 3D point $\mathbf{p} = [p_x, p_y, p_z]^\top \in \mathbb{R}^3$ can be rotated by treating it as a pure quaternion $[0, \mathbf{p}]$. The rotated point $\mathbf{p}'$ about an axis through the origin, specified by the unit quaternion $\mathbf{q}$, is obtained via quaternion multiplication:

$$[0, \mathbf{p}'] = \mathbf{q} \cdot [0, \mathbf{p}] \cdot \mathbf{q}^{-1}, \quad (2)$$

where $\mathbf{q}^{-1} = [w, -\mathbf{v}]$ denotes the conjugate of $\mathbf{q}$. Computationally, this rotation operation can be equivalently implemented by a $3 \times 3$ rotation matrix $R(\mathbf{q})$ derived directly form the components of $\mathbf{q}$:

$$R(\mathbf{q}) = \begin{bmatrix} 1 - 2(y^2 + z^2), 2(xy - zw), 2(xz + yw) \\ 2(xy + zw), 1 - 2(x^2 + z^2), 2(yz - xw) \\ 2(xz - yw), 2(yz + xw), 1 - 2(x^2 + y^2) \end{bmatrix}. \quad (3)$$

Thus, the rotation operation simplifies to the matrix multiplication $\mathbf{p}' = R(\mathbf{q})\mathbf{p}$. In contrast to Euler angles or rotation matrices, the quaternion representation not only avoids the gimbal lock but also supports smooth interpolation and stable optimization with only 4 parameters. These properties greatly reduce the model complexity and mitigate the risk of overfitting in deep learning.

The dot (or inner) product between two quaternions $\mathbf{q}_1 = [w_1, x_1, y_1, z_1]$ and $\mathbf{q}_2 = [w_2, x_2, y_2, z_2]$ is defined as:

$$\mathbf{q}_1 \cdot \mathbf{q}_2 = w_1 w_2 + x_1 x_2 + y_1 y_2 + z_1 z_2. \quad (4)$$

The dot product corresponds to the cosine of the angle between the rotations represented by the two quaternions, As such, it serves as an effective metric for evaluating geometric similarity in rotation space. This geometric property enables us to construct a rotation-aware similarity score, which forms the core of our proposed global attention mechanism.

## 3.2. QPoint Overview

The proposed QPoint is an end-to-end lightweight framework for point cloud processing based on quaternion representation. As illustrated in Figure 2, the pipeline begins by processing the input point cloud through Farthest Point Sampling (FPS) and K-Nearest Neighbor (KNN) algorithms to obtain local point features which are then mapped into a semantic feature space via Patch Embedding (PE), generating a set of tokenized representations. The resulting tokens are subsequently fed into a series of Quaternion Point Blocks (QPB), which enhance feature interaction by modeling both local geometric structures and global contextual relationships. Finally, the output features are directed to task-specific prediction heads to support various downstream applications.

### 3.2.1. PATCH EMBEDDING

Following established practices in point cloud processing (Pang et al., 2022), we begin by applying FPS to an input point cloud $\mathbf{P} \in \mathbb{R}^{N \times 3}$ containing $N$ points, yielding $L$ centroid points $\mathbf{P}_c \in \mathbb{R}^{L \times 3}$. For each centroid, we construct a local geometric patch by retrieving KNN from $\mathbf{P}$, resulting in a set of patches denotes as $\{\mathbf{x}_p^i \in \mathbb{R}^{K \times 3} \mid i = 1, 2, .., L\}$. These patches are then processed by a lightweight PointNet (Qi et al., 2017a) to perform patch embedding, which projects and compresses each patch into a $C$-dimensional feature representation, yielding the embedded patches set $\{x_p^i \in \mathbb{R}^{1 \times C} \mid i = 1, \ldots, L\}$. Following the Vision Transformer (ViT) paradigm (Dosovitskiy et al., 2020), we treat the $L$ patch embeddings as a token sequence. A learnable [CLS] token $x_{cls}^0 \in \mathbb{R}^{1 \times C}$ is prepended to this sequence to aggregate global sequence information, forming the initial token sequence $U_0$:

$$U_0 = \left[ x_{cls}^0; x_p^1; x_p^2; \cdots; x_p^L \right]. \tag{5}$$

The resulting sequence is processed by a stack of Quaternion Point Block (QPB) to learn hierarchical feature representations.

### 3.2.2. QPOINT BLOCK

As illustrated in Figure 2, each QPB consists of two core components: the Quaternion-Enhanced Local Perception (QELP) for local feature extraction and the Quaternion Global Attention (QGA) for global context modeling. Both components are preceded by Layer Normalization (LN) and employ residual connections. The computational flow of the $i$-th QPB is formalized as:

$$U_i^{\mathrm{m1}} = U_i^{\mathrm{in}} + X_{\mathrm{pos}}, \tag{6}$$

$$U_i^{\mathrm{m2}} = \mathrm{QELP}\left(\mathrm{LN}\left(U_i^{\mathrm{m1}}\right)\right) + U_i^{\mathrm{m1}}, \tag{7}$$

$$U_i^{\mathrm{out}} = \mathrm{QGA}\left(\mathrm{LN}\left(U_i^{\mathrm{m2}}\right)\right) + U_i^{\mathrm{m2}}, \tag{8}$$

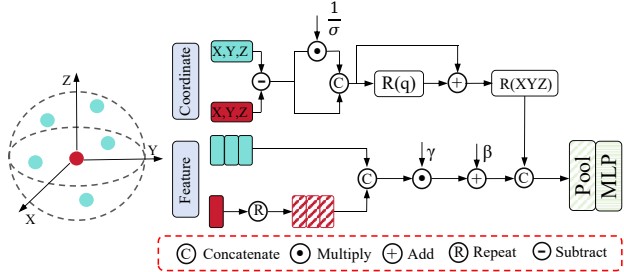

*Figure 3.* Quaternion-Enhanced Local Perception (QELP) normalizes local coordinates, applies learnable quaternion rotations, and calibrates features for robust geometric representation.

where $U_i^{\mathrm{in}}$ and $U_i^{\mathrm{out}}$ denote the input and output of the $i$-th QPB, respectively. The input to the first QPB is the patch-embedded sequence $U_0$ (i.e., $U_1^{\mathrm{in}} = U_0$). For subsequent blocks, the input equals the output of the preceding QPB (i.e., $U_i^{\mathrm{in}} = U_{i-1}^{\mathrm{out}}$ for $i > 1$). To enhance spatial awareness, we incorporate standard learnable position encoding $X_{\mathrm{pos}} \in \mathbb{R}^{(L+1) \times C}$ into the input of every QPB layer. In our implementation, We stack 4 blocks, and the output of final block is fed into a task-specific prediction head for various downstream tasks.

## 3.3. Quaternion-Enhanced Local Perception

The Quaternion-Enhanced Local Perception (QELP) enhances local geometric representation through a learnable quaternion rotation mechanism. By integrating coordinate normalization, feature expansion, and aggregation, QELP achieves rotation-robust feature extraction. The QELP framework is illustrated in Figure 3.

For the $i$-th local point group $\{\mathbf{p}_{i,j}\}_{j=1}^{K} \subset \mathbb{R}^3$ with $K$ neighborhood points, we first perform coordinate normalization to eliminate scale differences. The centroid (mean) coordinate $\boldsymbol{\mu}_i$ and standard deviation $\sigma_i$ of the group are computed as:

$$\boldsymbol{\mu}_i = \frac{1}{K} \sum_{j=1}^{K} \mathbf{p}_{i,j}, \quad \sigma_i = \sqrt{\frac{1}{K} \sum_{j=1}^{K} \|\mathbf{p}_{i,j} - \boldsymbol{\mu}_i\|^2 + \varepsilon}, \tag{9}$$

where $\varepsilon = 10^{-5}$ ensures numerical stability. Each neighborhood coordinate is normalized as $\hat{\mathbf{p}}_{i,j} = (\mathbf{p}_{i,j} - \boldsymbol{\mu}_i)/\sigma_i$, which facilitates stable rotation operations on a canonical spherical distribution.

To enhance adaptability to rotational transformations, we introduces a learnable quaternion rotation mechanism. Let $\mathbf{q} = [w, x, y, z]^\top \in \mathbb{R}^4$ be the learnable quaternion. During forward propagation, we first normalize $\mathbf{q}$ to ensure valid rotation, the unit quaternion corresponds to a 3D rotation matrix $R(\mathbf{q})$ (see Eq.3). The proposed module enhances input data through a dual-path process, operating on both spatial coordinates and feature representations. In the geometric enhancement stage, we apply a rotation to the normalized

coordinate $\hat{\mathbf{p}}_{i,j}$ and fuse it with the original via a residual connection:

$$\tilde{\mathbf{p}}_{i,j} = \hat{\mathbf{p}}_{i,j} + R(\mathbf{q})\,\hat{\mathbf{p}}_{i,j}. \tag{10}$$

This transformation, parameterized by a quaternion (only 4 learnable parameters), imbues the coordinate with rotational robustness at negligible computational cost. Concurrently, in the feature processing stage, we perform expansion and calibration. The centroid feature $\mathbf{f}_i^{\mathrm{ctr}}$ is duplicated and concatenated with each neighborhood point feature $\mathbf{f}_{i,j}$. This expanded feature is then dynamically calibrated using a per-channel affine transformation:

$$\mathbf{f}_{i,j}^{\mathrm{cat}} = [\mathbf{f}_{i,j}; \mathbf{f}_i^{\mathrm{ctr}}], \quad \mathbf{f}_{i,j}^{\mathrm{out}} = \boldsymbol{\gamma} \odot \mathbf{f}_{i,j}^{\mathrm{cat}} + \boldsymbol{\beta}, \tag{11}$$

where $\boldsymbol{\gamma}$ and $\boldsymbol{\beta}$ are learnable vectors that scale and shift the feature distribution in real time, respectively.

The output of QELP module is a joint tensor $\mathcal{G} = \{\tilde{\mathbf{p}}_{i,j}, \mathbf{f}_{i,j}^{\mathrm{out}}\}_{i=1,j=1}$, which integrates the rotation-enhanced coordinates with the calibrated features. This tensor is then passed through subsequent pooling and MLP layers to achieve dimension alignment. Through the collaborative enhancement of geometry and features, the QELP Block significantly improves the model's robustness to rotations while maintaining high computational efficiency.

## 3.4. Quaternion Global Attention

The Quaternion Global Attention (QGA) module effectively models global geometric relationships with inherent rotational awareness. It integrates quaternion representations within a self-attention framework to enable robust rotation-invariant feature interaction. The QLA framework is shown in figure 4.

### 3.4.1. QUATERNION ENCODING

Given an input feature tensor $\mathbf{X} \in \mathbb{R}^{N \times C}$, we first map it into a higher-dimensional space suitable for quaternion representation,

$$\mathbf{X}_{\mathrm{g}} = \mathrm{Linear}(\mathbf{X}) \in \mathbb{R}^{N \times (H \cdot d_k)}, \tag{12}$$

where $H$ denotes the number of attention heads and $d_k$ is the quaternion dimension per head. A critical constraint is that $H \cdot d_k$ must be a multiple of 4, ensuring the projected features can be partitioned into standard four-component quaternions of the form $[w, x, y, z]$. Each component is designed to encode a distinct aspect of geometric pose.

To preserve the rotational properties of quaternions, we apply unit normalization to the projected features:

$$\mathbf{X}_{\mathrm{unit}} = \frac{\mathbf{X}_{\mathrm{g}}}{\|\mathbf{X}_{\mathrm{g}}\|_2 + \epsilon}, \tag{13}$$

where $\|\cdot\|_2$ denotes the L2-norm and $\epsilon = 10^{-8}$ is a small constant for numerical stability.

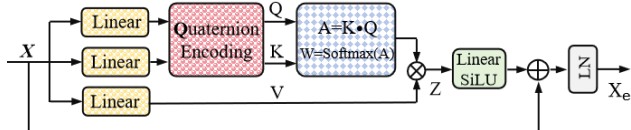

*Figure 4.* Quaternion Global Attention (QGA) uses quaternion dot products to compute rotation-aware attention scores for global context modeling.

### 3.4.2. ROTATION-AWARE ATTENTION MECHANISM

Building upon the quaternion-encoded features, we design a rotation-aware attention mechanism to model global feature correlations. The Query ($\mathbf{Q}$), key ($\mathbf{K}$), and value ($\mathbf{V}$) tensors are derived as follow:

$$\mathbf{Q} = W_q \mathbf{X}_{\mathrm{unit}}, \quad \mathbf{K} = W_k \mathbf{X}_{\mathrm{unit}}, \quad \mathbf{V} = W_v \mathbf{X}, \tag{14}$$

where $W_q, W_k, W_v$ are learnable projection matrices. Crucially, $\mathbf{Q}$ and $\mathbf{K}$ are computed from the unit-normalized quaternion feature $\mathbf{X}_{\mathrm{unit}}$ to preserve rotational geometry, while $\mathbf{V}$ is derived from the original feature $\mathbf{X}$ to retain semantic information.

The core of our method is the use of the quaternion dot product to compute attention scores. This measures the rotational alignment between features, a geometric relationship that standard inner products fail to capture:

$$\mathbf{A}_{ij} = \frac{\mathbf{Q}_i \cdot \mathbf{K}_j}{\sqrt{d_k}} = \frac{\sum_{c=1}^{d_k} \mathbf{Q}_{i,c} \mathbf{K}_{j,c}}{\sqrt{d_k}}, \tag{15}$$

where, $\mathbf{A}_{ij}$ is the attention score between the $i$-th and $j$-th features. The summation $\sum_{c=1}^{d_k} \mathbf{Q}_{i,c} \mathbf{K}_{j,c}$ explicitly computes the product sum for corresponding quaternion components (i.e., $w_i w_j + x_i x_j + y_i y_j + z_i z_j$), directly encoding their rotational similarity. The scaling factor $\sqrt{d_k}$ stabilizes gradients by preventing excessively large score magnitudes. The attention weights and output are then computed as:

$$\mathbf{W} = \mathrm{Softmax}(\mathbf{A}), \quad \mathbf{Z} = \mathbf{W}\mathbf{V}. \tag{16}$$

The output $\mathbf{Z}$ is a weighted sum of Values tensor, where the weights are determined by rotational relevance, allowing each feature to aggregate global context based on geometric alignment. Finally, to ensure training stability and dimension consistency, we employ a residual connections followed by layer normalization:

$$\mathbf{X}_{\mathrm{e}} = \mathrm{LN}(\mathbf{X} + \mathrm{SiLU}(\mathrm{Linear}(\mathbf{Z}))), \tag{17}$$

where LN denotes layer normalization (LayerNorm) and SiLU is an activation function. The enhanced feature $\mathbf{X}_{\mathrm{e}} \in \mathbb{R}^{N \times C}$ integrates rotation-aware global context while preserving orignal local geometric information, making it highly effective for downstream point cloud tasks like classification and segmentation. More theoretical analysis of our method are provided in the *Appendix*.

*Table 1.* Classification on the ScanObjectNN and ModelNet40. We report the overall accuracy (OA, %) on three variants of ScanObjectNN (OBJ_BG, OBJ_ONLY, and PB_T50_RS), the OA and mean accuracy (mAcc, %) on ModelNet40, the parameters (M) and FLOPs (G) on both ScanObjectNN and ModelNet40.

| Method | ScanObjectNN | | | ModelNet40 | | Params(M) ↓ | FLOPs(G) ↓ |
| | OBJ_BG (OA) ↑ | OBJ_ONLY (OA) ↑ | PB_T50_RS (OA) ↑ | OA ↑ | mAcc ↑ | | |
|---|---|---|---|---|---|---|---|
| *MLP/CNN-based* | | | | | | | |
| PointNet++ (Qi et al., 2017b) | 82.3 | 84.3 | 77.9 | 91.9 | - | 1.5 | 1.7 |
| MVTN (Hamdi et al., 2021) | 92.6 | 92.3 | 82.8 | 93.8 | - | 11.2 | 43.7 |
| PointMLP (Ma et al., 2022) | - | - | 85.4 | 94.1 | 91.3 | 12.6 | 31.4 |
| PointNeXt (Qian et al., 2022) | - | - | 87.7 | 93.2 | 90.8 | **1.4** | 3.6 |
| *Mamba-based* | | | | | | | |
| PointMamba (Liang et al., 2024) | 90.7 | 88.4 | 84.8 | 93.6 | - | 12.3 | 3.6 |
| Mamba3D (Han et al., 2024) | 92.9 | 92.1 | **91.8** | 93.4 | - | 16.9 | 3.9 |
| PCM (Zhang et al., 2025) | - | - | 88.1 | 93.4 | - | 34.2 | 45.0 |
| SIM (Bahri et al., 2025) | 92.3 | 91.4 | 87.3 | 92.7 | - | 12.3 | 3.6 |
| *Transformer-based* | | | | | | | |
| PointTransformer (Zhao et al., 2021) | - | - | - | 93.7 | 90.6 | 22.1 | 4.8 |
| Point-BERT (Yu et al., 2022) | 87.4 | 88.1 | 83.1 | 93.2 | - | 22.1 | 4.8 |
| Point-MAE (Pang et al., 2022) | 90.0 | 88.3 | 85.2 | 93.8 | - | 22.1 | 4.8 |
| Point-M2AE (Zhang et al., 2022) | 91.2 | 88.8 | 86.4 | 94.0 | - | 15.3 | 3.6 |
| **QPoint** *w/o voting.* | **93.8** | **92.6** | 90.7 | **94.2** | **91.4** | 4.5 | **0.9** |
| **QPoint** *w/ voting.* | **95.0** | **93.9** | **92.1** | **94.7** | **91.6** | 4.5 | **0.9** |

# 4. Experiments

We evaluate QPoint on several standard benchmarks, including ModelNet40 (Wu et al., 2015) and the three variants of ScanObjectNN (Uy et al., 2019) (PB_T50_RS, OBJ_BG, OBJ_ONLY) for classification, as well as ShapeNet (Yi et al., 2016) for part segmentation. All experiments are conducted on a server with an AMD Ryzen Threadripper 3960X 3.80GHz 24-core processor and an NVIDIA A6000 GPU. Comprehensive dataset statistics and implementation details are provided in the *Appendix*.

## 4.1. Comparison with State-of-the-art Methods

We conduct comprehensive comparisons between QPoint and existing state-of-the-art methods across multiple point cloud tasks. For classification, We report Overall Accuracy (OA) on ScanObjectNN, while both OA and mean Accuracy (mAcc) on ModelNet40. Model complexity is assessed via parameter count (Params) and FLOPs for all classification setups. In the few-shot setting on ModelNet40, we report mAcc with standard deviation. For part segmentation on ShapeNetPart, we adopt mean Intersection-over-Union per class (Cls.mIoU) and per instance (Inst.mIoU), along with Parameters (Params) and FLOPs.

**Classification on ScanObjectNN.** ScanObjectNN (Uy et al., 2019) is a challenging real-world 3D object dataset collected from indoor scenes, containing around 15,000 objects across 15 categories with cluttered background and occlusions. We evaluate QPoint on three variants a training-from-scratch setting. As summarized in Table 1, on OBJ_BG, QPoint achieves 93.8% OA, and its maximum OA rose to 95.0% after optimization with voting, surpassing Mamba3D by 2.1%. On OBJ_ONLY, QPoint has an OA of 92.6%, with a maximum OA of 93.9% after voting, outper-

form MVTN by 1.6%. Even on the most challenging variant PB_T50_RS, QPoint achieved an OA of 90.7%, and its maximum OA reached 92.1% after voting, exceeding Mamba3D by 0.3%. *Notably, it attains this this superior performance with only **26.6%** of the parameters and **25%** of the computational cost (FLOPs) of Mamba3D*. QPoint achieves current state-of-the-art performance with a lightweight network architecture, and the results are convincing.

**Classification on ModelNet40.** ModelNet40 (Wu et al., 2015) is widely recognized benchmark for 3D object classification, covering 40 common object categories with a total of 12,311 synthetic CAD models and a standard train/test split of 9843/2468 instances. As reported in Table 1, QPoint attains 94.2% OA when trained from scratch without any pre-training or external data. With voting strategy (Liu et al., 2019) applying during inference, the overall accuracy further improves to 94.7%. This result surpasses Mamba3D (93.4%) by 1.3% and also outperforms the recent state-of-the-art Point-M2AE (94.0%) by 0.7%. Notably, QPoint achieves this leading performance with only 4.5M learnable parameters and 0.9G FLOPs, fully demonstrating a highly favorable accuracy-efficiency trade-off compared to existing Transformer-based or Mamba-based models.

**Part Segmentation on ShapeNetPart.** ShapeNetPart (Yi et al., 2016) is a canonical benchmark for fine-grained part segmentation, comprising 16 shape categories with a total of 16,880 3D models, each annotated with up to 50 part labels. As reported in Table 2, QPoint achieves strong performance on this task, reaching 85.1% class mIoU and 87.0% instance mIoU with only 8.8M parameters and 4.3G FLOPs. This performance is on par with the current best SOTA Mamba-based method PCM (84.8% Cls.mIoU) and surpasses the Transformer-based Point-M2AE (86.5% Inst.mIoU), *which requiring only **21.6%** of the parameters and **9%** of the*

*Table 2.* Part segmentation on ShapeNetPart. The mean Intersection-over-Union per class (Cls. mIoU, %) and per instance (Inst. mIoU, %) are reported, along with Params (M) and FLOPs (G).

| Method | Backbone | Cls. mIoU ↑ | Inst. mIoU ↑ | Params ↓ | FLOPs ↓ |
|---|---|---|---|---|---|
| PointNet++ (Qi et al., 2017b) | MLP-based | 81.8 | 85.1 | **4.0** | 4.9 |
| APES (Wu et al., 2023) | MLP-based | 83.6 | 85.8 | - | - |
| PointMLP (Ma et al., 2022) | MLP-based | 84.6 | 86.1 | - | - |
| PointNeXt (Qian et al., 2022) | MLP-based | 85.2 | 87.0 | 22.5 | 110.2 |
| PointMamba (Liang et al., 2024) | Mamba-based | 84.4 | 86.0 | 17.4 | 14.3 |
| Mamba3D (Han et al., 2024) | Mamba-based | 84.1 | 85.7 | 21.9 | 9.5 |
| PCM (Zhang et al., 2025) | Mamba-based | **85.3** | **87.0** | 40.6 | 45.0 |
| SIM (Bahri et al., 2025) | Mamba-based | 84.1 | 85.9 | 12.3 | 14.3 |
| PointTransformer (Zhao et al., 2021) | Transformer-based | 83.7 | 86.5 | - | - |
| Point-BERT (Yu et al., 2022) | Transformer-based | 84.1 | 85.6 | 27.1 | 10.6 |
| Point-MAE (Pang et al., 2022) | Transformer-based | 84.2 | 86.1 | 27.1 | 15.5 |
| Point-M2AE (Zhang et al., 2022) | Transformer-based | 84.8 | 86.5 | - | - |
| **QPoint(ours)** | Transformer-based | 85.1 | **87.0** | 8.8 | **4.3** |

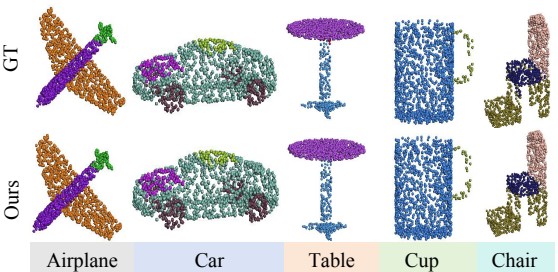

GT / Ours

Airplane | Car | Table | Cup | Chair

*Figure 5.* Part segmentation results on ShapeNetPart. Top row is ground truth and bottom row is our prediction.

*FLOPs of PCM.* These results fully demonstrate QPoint's ability to balance high accuracy with lightweight design. In addition, Qualitative results in Figure 5 further confirm that QPoint produces segmentation outputs nearly identical to the ground truth, underscoring its visual reliability and geometric awareness.

**Few-shot Classification on ModelNet40.** To further access QPoint's capability under data scarcity, we conduct a comprehensive few-shot classification evaluation on ModelNet40. The experiments strictly follow the n-way, m-shot protocol from (Sharma & Kaul, 2020), examining four combinations of $n \in \{5, 10\}$ and $m \in \{10, 20\}$, covering the extremely sparse range of labeled samples from $5 \times 10 = 50$ to $10 \times 20 = 200$. As summarized in Table 3. QPoint trained from scratch achieves OA scores of 95.0%, 98.1%, 90.7%, and 94.1% across these settings, outperforming the best existing methods by margins of 2.4%, 1.2%, 2.6%, and 1.0%, respectively. Notably, QPoint also exhibits significantly lower standard deviation, confirming more stable and stable convergence. These results strongly validate QPoint's effectiveness in knowledge transfer and generalization for data-constrained point cloud understanding.

**Semantic segmentation on S3DIS dataset.** On the large-scale S3DIS dataset, QPoint achieves an optimal balance between lightweight design and high performance, which is critical for practical deployment. As shown in Table 4,

*Table 3.* Few-shot classification on ModelNet40. We report the average accuracy (%) and standard deviation (%) with only supervised learning.

| Method | 5-way | | 10-way | |
|---|---|---|---|---|
| | 10-shot ↑ | 20-shot ↑ | 10-shot ↑ | 20-shot ↑ |
| PointNet (Qi et al., 2017a) | 52.0 ± 3.8 | 57.8 ± 4.9 | 46.6 ± 4.3 | 35.2 ± 4.8 |
| PointNet++ (Qi et al., 2017b) | 38.5 ± 3.6 | 42.4 ± 2.9 | 23.1 ± 4.8 | 18.8 ± 5.2 |
| DGCNN (Wang et al., 2019) | 91.8 ± 3.7 | 93.4 ± 3.2 | 86.3 ± 6.2 | 90.9 ± 5.1 |
| HyMamba (Liu et al., 2025) | 90.5 ± 3.8 | 96.0 ± 3.2 | 86.3 ± 5.1 | 92.0 ± 3.4 |
| Mamba3D (Han et al., 2024) | 92.6 ± 3.7 | 96.9 ± 2.4 | 88.1 ± 5.3 | 93.1 ± 3.6 |
| **QPoint(ours)** | **95.0 ± 3.4** | **98.1 ± 1.7** | **90.7 ± 5.0** | **94.1 ± 2.3** |

QPoint attains a competitive mIoU of 71.2%, closely matching the performance of heavier models like PTv2 (71.6%) and PTv3 (73.6%). Its key advantage, however, lies in its exceptional efficiency: with only 8.8M parameters and 9.4G FLOPs, it drastically reduces computational costs compared to these counterparts (e.g., PTv3 at 46.2M Params/149.3G FLOPs). This demonstrates that QPoint provides a highly practical solution for resource-constrained scenarios like edge deployment, offering high accuracy with minimal resource consumption. In addition, We evaluate our method on S3DIS Area 5, a standard benchmark for large-scale indoor scene segmentation. As shown in Figure 6, our predictions (bottom) closely align with the ground truth (top) across various scenes. The model accurately delineates semantic boundaries between objects like walls, floors, and furniture, even in complex layouts. This high consistency with GT annotations demonstrates robust performance in understanding diverse indoor environments.

### 4.2. Ablation Studies

We conduct comprehensive ablation studies on ModelNet40 or ScanObjectNN dataset to evaluate the contribution of each component in QPoint. Our experiments assess several key factors: robustness to perturbations, token length, number of block and the effectiveness of individual components. More experiments, including neighbor scale, point scale, qualitative visualizations are provided in the *Appendix*.

*Table 4.* Semantic Segmentation Results on S3DIS Dataset.

| Methods | mIoU | ceiling | floor | wall | beam | column | window | door | table | chair | sofa | bookcase | board | clutter | Params | FLOPs |
|---|---|---|---|---|---|---|---|---|---|---|---|---|---|---|---|---|
| PointNet (Qi et al., 2017a) | 41.1 | 88.8 | 97.3 | 69.8 | 0.0 | 3.9 | 46.3 | 10.8 | 59.0 | 52.6 | 5.9 | 40.3 | 26.4 | 33.2 | 3.6 | 4.9 |
| PointNet++ (Qi et al., 2017b) | 53.5 | 89.4 | 97.7 | 75.4 | 0.0 | 1.8 | 58.3 | 19.5 | 79.0 | 69.2 | 59.1 | 46.2 | 58.7 | 41.6 | 4.0 | 4.9 |
| PointCNN (Li et al., 2018) | 57.3 | 92.3 | 98.2 | 79.4 | 0.0 | 17.6 | 22.8 | 62.1 | 74.4 | 80.6 | 31.7 | 66.7 | 62.1 | 56.7 | 46.2 | 70.5 |
| PointNeXt (Qian et al., 2022) | 70.5 | 94.2 | 98.5 | 84.4 | 0.0 | 37.7 | 59.3 | 74.0 | 83.1 | 91.6 | 77.4 | 77.2 | 78.8 | 60.6 | - | - |
| PCM (Zhang et al., 2024) | 63.4 | 93.3 | 96.7 | 80.6 | 0.0 | 35.9 | 57.7 | 60.0 | 74.0 | 87.6 | 50.1 | 69.4 | 63.5 | 55.9 | - | - |
| PointRWKV (He et al., 2024) | 70.5 | 94.2 | 98.3 | 86.5 | 0.0 | 38.6 | 64.5 | 76.2 | 88.2 | 89.3 | 65.2 | 75.6 | 78.2 | 61.3 | - | - |
| E-3DSNN (Qiu et al., 2025) | 67.4 | 95.3 | 98.5 | 82.3 | 28.0 | 55.8 | 71.5 | 81.2 | 89.8 | 69.2 | 76.4 | 67.0 | 61.6 | - | 14.4 | |
| 3DSMT (Zhou et al., 2026) | 70.2 | 88.9 | 94.2 | 82.5 | 46.8 | 62.0 | 74.4 | 85.3 | 87.3 | 77.3 | 76.9 | 77.6 | 59.8 | - | 11.4 | |
| PTv1 (Zhao et al., 2021) | 70.4 | 94.0 | 98.5 | 86.3 | 0.0 | 38.0 | 63.4 | 74.3 | 89.1 | 82.4 | 74.3 | 80.2 | 76.0 | 59.3 | 4.9 | 16.7 |
| PTv2 (Wu et al., 2022) | 71.6 | 93.0 | 98.1 | 86.7 | 0.0 | 48.0 | 62.4 | 76.1 | 88.3 | 87.6 | 77.1 | 79.2 | 77.5 | 59.8 | 12.8 | 86.9 |
| PTv3 (Wu et al., 2024) | 73.6 | 92.4 | 98.3 | 86.6 | 0.0 | 55.8 | 63.7 | 77.1 | 83.8 | 93.3 | 79.1 | 79.4 | 85.4 | 61.7 | 46.2 | 149.3 |
| QPoint(ours) | **71.2** | 90.6 | 96.9 | 84.8 | 0.0 | 48.9 | 61.8 | 75.6 | 86.8 | 89.2 | 77.6 | 78.4 | 77.9 | 59.9 | 9.8 | 9.4 |

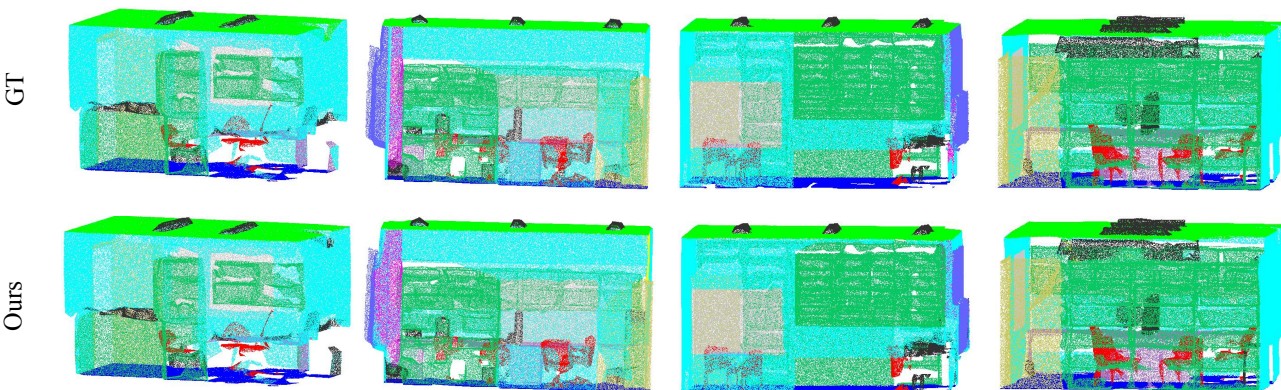

*Figure 6.* Semantic segmentation results on S3DIS. Top row is ground truth and bottom row is our prediction.

*Table 5.* Model Efficiency on ModelNet40 (FLOPs, Params, Latency, Memory).

| Methods | Training | | Inference | |
|---|---|---|---|---|
| | **Latency** | **Memory** | **Latency** | **Memory** |
| SPT-512(Wu et al., 2025) | 326ms | 9.7G | 191ms | 5.2G |
| SPT-768(Wu et al., 2025) | 385ms | 12.5G | 201ms | 7.3G |
| SPT-1024(Wu et al., 2025) | 431ms | 15.2G | 227ms | 9.5G |
| PointMLP (Ma et al., 2022) | 515ms | 18.7G | 313ms | 12.9G |
| Mamba3D (Han et al., 2024) | 433ms | 14.9G | 256ms | 7.8G |
| QPoint | **142ms** | **3.1G** | **86ms** | **1.4G** |

**Running Efficiency Analysis.**

To evaluate runtime efficiency, we compare QPoint against Mamba3D, PointMLP and SPT variants on ModelNet40. As shown in Table 5, QPoint demonstrates superior efficiency across all metrics. QPoint's training latency (142ms) and GPU memory usage (3.1G) are less than one-third of Mamba3D's, and far lower than other compared methods. During inference, it maintains this prominent advantage with a latency of 86ms and memory use of 1.4G. These results confirm that QPoint achieves comprehensive efficiency optimization, making it highly suitable for real-time and resource-constrained applications.

**Analysis of Robustness.** To evaluate model robustness under under complex geometric transformations, we conduct

comprehensive robustness experiments during the testing by examining resistance to Y-axis rotations, Z-axis translations, scaling, and point cloud jitter. As shown in Table 6, under the baseline without any transformations (None), QPoint reached an OA of 94.2%. In Y-axis rotation tests (-90°, 90°, 180°), while competing methods suffer significant performance degradation, QPoint maintains a robust OA of 80.6%, which we attribute to the use of quaternions in mitigating rotational distortion. Under Z-axis translation (±0.2 units), QPoint achieves the highest OA of 93.7%, demonstrating low sensitivity to spatial shifts. In scaling tests (ranging from 0.5-1.5 to 0.9-1.1), QPoint consistently sustains stable performance (93.7% OA), with its advantage becoming more pronounced under narrower scaling ranges. Furthermore, under point cloud jitter, QPoint outperforms other methods with 92.7% OA, highlighting its strong resistance to local perturbations. We also conducted robustness tests on each module of QPoint. When we removed the QELP module and QGA module separately, QPoint's accuracy slightly decreased in all 3D robustness transformation tests. But each module has a certain degree of rotational invariance. These results collectively validate the high robustness of QPoint against diverse 3D geometric transformations.

**Effect of Different Components.** To evaluate the contribution of each core module in QPoint, we conduct an

*Table 6.* Robustness to Rigid Transformations Around Y-Axis, Translation Along Z-Axis, Scaling and Jittering on ModelNet40.

| Method | None | Rotation | | | Translation | | Scaling | | | | | Jittering |
|---|---|---|---|---|---|---|---|---|---|---|---|---|
| | | -90° | 90° | 180° | +0.2 | -0.2 | 0.5-1.5 | 0.6-1.4 | 0.7-1.3 | 0.8-1.2 | 0.9-1.1 | |
| PointNet++ (Qi et al., 2017b) | 91.9 | 57.9 | 57.9 | 57.9 | 90.7 | 90.8 | 91.2 | 91.2 | 90.9 | 91.0 | 91.0 | 89.6 |
| DGCNN (Wang et al., 2019) | 92.9 | 55.6 | 56.5 | 74.0 | 92.3 | 92.3 | 90.7 | 91.6 | 92.1 | 92.3 | 91.8 | 91.5 |
| DANet (He et al., 2023) | 93.0 | 58.7 | 59.3 | 72.9 | 92.6 | 92.7 | 92.3 | 92.4 | 92.3 | 92.6 | 92.6 | 91.6 |
| Mamba3D (Han et al., 2024) | 93.4 | 59.4 | 60.1 | 60.1 | 92.9 | 92.9 | 92.6 | 92.7 | 92.6 | 93.0 | 93.0 | 92.0 |
| PTv3 (Wu et al., 2024) | 94.2 | 60.7 | 61.2 | 75.1 | 93.5 | 93.6 | 93.3 | 93.2 | 93.2 | 93.7 | 93.6 | 92.6 |
| QPoint (Full) | **94.2** | **80.5** | **80.6** | **80.6** | **93.7** | **93.7** | **93.5** | **93.5** | **93.3** | **93.7** | **93.7** | **92.7** |
| QPoint (no/QELP) | **93.0** | **75.3** | **75.3** | **75.3** | **92.7** | **92.7** | **92.4** | **92.5** | **92.4** | **92.8** | **92.8** | **92.0** |
| QPoint (no/QGA) | **92.7** | **72.6** | **72.6** | **72.6** | **92.1** | **92.1** | **91.8** | **92.0** | **91.9** | **92.3** | **92.3** | **91.5** |

*Table 7.* Ablation study results on ScanObjectNN and ModelNet40, evaluating the effectiveness of the QELP and QGA modules.

| Method | ScanObjectNN | | | ModelNet40 |
|---|---|---|---|---|
| | OBJ-BG ↑ | OBJ-ONLY ↑ | PB-T50-RS ↑ | OA (%) ↑ |
| Full | **93.8** | **92.6** | **90.7** | **94.2** |
| no/ QELP-Q | 93.1 | 92.0 | 89.8 | 93.6 |
| no/ QELP | 92.3 | 91.6 | 89.2 | 93.0 |
| no/ QGA-Q | 92.6 | 91.8 | 89.6 | 93.1 |
| no/ QGA | 91.9 | 91.2 | 88.9 | 92.7 |

*Table 8.* Effect of QPB number on ModelNet40

| $Num$ | OA (%) | mAcc (%) |
|---|---|---|
| 1 | 93.5 | 90.3 |
| 2 | 93.8 | 90.7 |
| 4 | **94.2** | **91.4** |
| 6 | 94.0 | 91.1 |
| 8 | 93.9 | 91.0 |

*Table 9.* Effect of $L$ tokens on ModelNet40

| $L$ | OA (%) | mAcc (%) |
|---|---|---|
| 32 | 92.6 | 89.6 |
| 64 | 93.4 | 90.2 |
| 128 | **94.2** | **91.4** |
| 256 | 94.0 | 90.9 |
| 384 | 93.5 | 90.4 |

ablation study comparing different schemes' performance on ScanObjectNN and ModelNet40 (Table 7). In our experimental setup, "QELP-Q" refers to the QELP module without learnable quaternion rotation coordinates, while "QGA-Q" denotes the QGA module deprived of quaternion-based rotation-aware similarity scoring. Compared to the full model, removing the quaternion component in QELP (i.e., QELP–Q) leads to a slight but consistent drop across all evaluation metrics, attesting to its role in enhancing local feature representation. A more pronounced performance degradation is observed when the entire QELP module is removed, underscoring its importance as the cornerstone of local feature encoding. Similarly, ablating the quaternion-guided scoring in QGA (QGA–Q) results in a noticeable decline in accuracy, validating its effectiveness in modeling global contextual relationships. The most severe performance drop occurs when the complete QGA module is excluded, highlighting its indispensability for global feature aggregation. These findings collectively affirm the synergistic contribution of both QELP and QGA modules to the overall robustness and discriminative power of QPoint.

**Effect of QPB Number.** QPoint comprises multiple stacked Quaternion Point Blocks (QPBs). To determine the optimal depth, we conduct ablation studies on the ModelNet40 by stacking 1, 2, 4, 6, and 8 QPBs. As shown in Table 8, classificaiton accuracy improves with additional QPBs, peaking at 94.2% OA with 4 blocks. The QPoint effectively captures complex inter-point relationships, enhancing discriminative capability. However, using 8 QPBs introduces excessive computational burden and information redundancy, leading to slightly degraded performance.

**Effect of Token Length.** Similar to ViT (Dosovitskiy et al., 2020), we investigate the impact of token length $L$ on QPoint performance through ablation studies on ModelNet40, As shown in Table 9, when $L$ increases from 32 to 128, both OA and mAcc consistently improve, reaching peak performance at $L$=128 (OA **94.2%**, mAcc **91.4%**). Further increasing $L$ to 256 leads to a slight decline in both metrics (OA 94.0%, mAcc 90.9%). These results indicate that $L$=128 achieves the optimal trade-off between representation capacity and learning efficiency.

## 5. Conclusion

We propose QPoint, an end-to-end lightweight framework that incorporates quaternion geometric priors to address point cloud rotational sensitivity and efficiency constraints. Its key innovations include: the Quaternion-Enhanced Local Perception module, enhancing rotational robustness through learnable quaternion rotations; and the Quaternion Global Attention module, achieving rotation-aware global context modeling via quaternion dot products. Comprehensive experiments demonstrate that QPoint's exceptional balance of efficiency and performance. On many benchmarks, our method achieves state-of-the-art results in classification, segmentation, and few-shot learning tasks, while substantially reducing parameters and computation. This confirms the great potential of quaternions for building efficient, robust 3D vision models. QPoint's success illustrates that integrating mathematical tools like quaternions with deep learning effectively addresses fundamental 3D geometric learning challenges. Future work will extend this paradigm to tasks such as point cloud registration and generation.

## Acknowledgments

This research was partially supported by the Australian Research Council Discovery Project (DP240101926) and the National Natural Science Foundation of China (Nos. 52575580, 62236002, U25A20533).

## Impact Statement

This paper presents work whose goal is to advance the field of Machine Learning. There are many potential societal consequences of our work, none which we feel must be specifically highlighted here.

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

**The supplementary material presents the following sections to strengthen the main manuscript:**

**A1.** Theoretical Analysis.

**A2.** Dataset Introduction.

**A3.** Experiment Details.

**A4.** More Experiment.

**A5.** More Visualization.

## A1. Theoretical Analysis

### A1.1. Quaternion Representation and SO(3) Group Homomorphism

Unit quaternions form a group under multiplication that is isomorphic to the special unitary group SU(2). The mapping $\phi : S^3 \to$ SO(3) from unit quaternions to 3D rotation matrices is a two-to-one surjective homomorphism defined by:

$$\phi(\mathbf{q}) = \mathbf{R}(\mathbf{q}) = \begin{bmatrix} 1 - 2(y^2 + z^2) & 2(xy - zw) & 2(xz + yw) \\ 2(xy + zw) & 1 - 2(x^2 + z^2) & 2(yz - xw) \\ 2(xz - yw) & 2(yz + xw) & 1 - 2(x^2 + y^2) \end{bmatrix}. \tag{18}$$

This mapping satisfies $\phi(\mathbf{q}_1\mathbf{q}_2) = \phi(\mathbf{q}_1)\phi(\mathbf{q}_2)$ and $\phi(-\mathbf{q}) = \phi(\mathbf{q})$, establishing the well-known double cover of SO(3) by SU(2). This representation provides a singularity-free and compact parameterization of 3D rotations with only 4 parameters, avoiding the gimbal lock problem inherent in Euler angles.

### A1.2. Theoretical Analysis of Rotational Robustness

We distinguish between two types of rotational robustness in QPoint: the adaptive stabilization provided by QELP and the strict invariance achieved by QGA.

#### A1.2.1. QELP: ADAPTIVE STABILIZATION VIA LEARNED ROTATION CORRECTION

The QELP module does not provide strict equivariance but rather achieves adaptive stabilization against input rotations through a learnable rotation mechanism in a normalized local coordinate frame. Let $\mathcal{P} = \{\mathbf{p}_i\}$ be a local point set and $\mathcal{P}' = \{\mathbf{R}\mathbf{p}_i + \mathbf{t}\}$ be its transformed version under rotation $\mathbf{R}$ and translation $\mathbf{t}$. QELP first performs normalization:

$$\hat{\mathbf{p}}_i = \frac{\mathbf{p}_i - \boldsymbol{\mu}}{\sigma}, \quad \boldsymbol{\mu} = \frac{1}{K} \sum_i \mathbf{p}_i, \quad \sigma = \sqrt{\frac{1}{K} \sum_i \|\mathbf{p}_i - \boldsymbol{\mu}\|^2}. \tag{19}$$

For the transformed set, we have $\boldsymbol{\mu}' = \mathbf{R}\boldsymbol{\mu} + \mathbf{t}$ and $\sigma' = \sigma$ (since rotation preserves distances). The normalized coordinates become:

$$\hat{\mathbf{p}}'_i = \frac{\mathbf{R}\mathbf{p}_i + \mathbf{t} - (\mathbf{R}\boldsymbol{\mu} + \mathbf{t})}{\sigma} = \mathbf{R}\hat{\mathbf{p}}_i. \tag{20}$$

Thus, normalization eliminates translation and isolates the rotation in a canonical coordinate frame centered at the local centroid. The subsequent learnable rotation $\mathbf{R}(\mathbf{q})$ operates in this normalized frame. The enhanced coordinates are computed as:

$$\tilde{\mathbf{p}}'_i = \hat{\mathbf{p}}'_i + \mathbf{R}(\mathbf{q})\hat{\mathbf{p}}'_i = \mathbf{R}\hat{\mathbf{p}}_i + \mathbf{R}(\mathbf{q})(\mathbf{R}\hat{\mathbf{p}}_i). \tag{21}$$

Importantly, since matrix multiplication is not commutative ($\mathbf{R}(\mathbf{q})\mathbf{R} \neq \mathbf{R}\mathbf{R}(\mathbf{q})$ unless they share the same axis), we cannot extract a common factor of $\mathbf{R}$. Therefore, QELP does not provide strict equivariance. Instead, the learnable quaternion $\mathbf{q}$ is optimized through gradient descent to adaptively rotate the normalized local structure to a "canonical pose" that minimizes the downstream task loss. This mechanism enables the network to learn a function $f$ that satisfies:

$$f\left(\mathbf{R}\hat{\mathbf{p}}_i + \mathbf{R}(\mathbf{q})(\mathbf{R}\hat{\mathbf{p}}_i)\right) \approx f\left(\hat{\mathbf{p}}_i + \mathbf{R}(\mathbf{q}_0)\hat{\mathbf{p}}_i\right) \tag{22}$$

for different rotations $\mathbf{R}$, where $\mathbf{q}_0$ represents an optimal baseline rotation. This adaptive stabilization significantly reduces sensitivity to input rotations while maintaining minimal parameter overhead (only 4 parameters).

A1.2.2. QGA: STRICT ROTATIONAL INVARIANCE VIA QUATERNION DOT PRODUCT

The QGA module achieves strict rotational invariance in global feature interaction. Let $\mathbf{X} \in \mathbb{R}^{N \times C}$ be input features. After quaternion encoding and unit normalization, we obtain quaternion features $\mathbf{Q}, \mathbf{K} \in \mathbb{R}^{N \times d_k}$ where $d_k$ is a multiple of 4. Consider a global rotation $\mathbf{R}$ applied to the input point cloud. In the quaternion domain, this corresponds to left-multiplication by a unit quaternion $\mathbf{r}$ representing $\mathbf{R}$. The transformed quaternion features become $\mathbf{Q}' = \mathbf{r} \otimes \mathbf{Q}$ and $\mathbf{K}' = \mathbf{r} \otimes \mathbf{K}$, where $\otimes$ denotes element-wise quaternion multiplication. The quaternion dot product between transformed features is:

$$\mathbf{Q}'_i \cdot \mathbf{K}'_j = (\mathbf{r} \otimes \mathbf{q}_i) \cdot (\mathbf{r} \otimes \mathbf{k}_j). \tag{23}$$

Expanding using the definition $\mathbf{a} \cdot \mathbf{b} = \mathrm{Re}(\mathbf{a}\mathbf{b}^*)$:

$$(\mathbf{r} \otimes \mathbf{q}_i) \cdot (\mathbf{r} \otimes \mathbf{k}_j) = \mathrm{Re}\left((\mathbf{r}\mathbf{q}_i)(\mathbf{r}\mathbf{k}_j)^*\right) \tag{24}$$

$$= \mathrm{Re}\left(\mathbf{r}\mathbf{q}_i\mathbf{k}_j^*\mathbf{r}^*\right) \tag{25}$$

$$= \mathrm{Re}\left(\mathbf{q}_i\mathbf{k}_j^*\right) \quad \text{(since } \mathbf{r}^* = \mathbf{r}^{-1} \text{ and real part is invariant under conjugation)} \tag{26}$$

$$= \mathbf{q}_i \cdot \mathbf{k}_j. \tag{27}$$

Thus, the attention scores $\mathbf{A}_{ij} = \frac{\mathbf{Q}_i \cdot \mathbf{K}_j}{\sqrt{d_k}}$ are invariant to global rotations. Since the value matrix $\mathbf{V}$ is derived from the original features (not the quaternion-normalized ones), the output $\mathbf{Z} = \mathbf{W}\mathbf{V}$ inherits this invariance, providing strict rotational invariance in global context modeling.

**A1.3. Geometric Interpretation of Quaternion Dot Product**

The quaternion dot product $\mathbf{q}_1 \cdot \mathbf{q}_2$ has a deep geometric interpretation. Let $\mathbf{q}_1 = [\cos(\theta_1/2), \sin(\theta_1/2)\mathbf{u}_1]$ and $\mathbf{q}_2 = [\cos(\theta_2/2), \sin(\theta_2/2)\mathbf{u}_2]$ represent rotations by angles $\theta_1, \theta_2$ around axes $\mathbf{u}_1, \mathbf{u}_2$ (unit vectors). Their dot product expands to:

$$\mathbf{q}_1 \cdot \mathbf{q}_2 = \cos\left(\frac{\theta_1}{2}\right)\cos\left(\frac{\theta_2}{2}\right) + \sin\left(\frac{\theta_1}{2}\right)\sin\left(\frac{\theta_2}{2}\right)\mathbf{u}_1 \cdot \mathbf{u}_2. \tag{28}$$

When $\mathbf{u}_1 = \mathbf{u}_2$, this simplifies to $\cos((\theta_1 - \theta_2)/2)$, directly measuring the angular difference between rotations. In general, it measures the cosine of the half-angle between the two rotations in the rotation space. This geometric property enables QGA to compute attention weights based on rotational alignment, inherently focusing on geometrically relevant features regardless of global orientation.

**A1.4. Quantitative Bound on Rotational Invariance**

To quantify the rotational robustness of QPoint, we define the invariance error for a model $\mathcal{M}$ under random rotations:

$$\mathcal{E}_{\mathrm{inv}} = \mathbb{E}_{\mathbf{R} \sim \mathrm{Uniform}(SO(3))}\left[\|\mathcal{M}(\mathbf{R}\mathcal{P}) - \mathcal{M}(\mathcal{P})\|^2\right]. \tag{29}$$

For QGA, we have proven $\mathcal{E}_{\mathrm{inv}} = 0$ due to strict invariance. For QELP, while not strictly invariant, the adaptive stabilization minimizes this error through learning. Empirically, as shown in Table 6, QPoint maintains 80.6% accuracy under $180°$ Y-axis rotations, a drop of only 13.6% from the non-rotated baseline (94.2%). In contrast, PointNet++ suffers a 34.0% drop (91.9% $\rightarrow$ 57.9%). This demonstrates that QPoint's combination of adaptive stabilization (QELP) and strict invariance (QGA) achieves significantly better robustness than methods lacking explicit geometric priors.

*Table A1.* Experimental settings.

| Classification | | Segmentation | |
|---|---|---|---|
| **Config** | **Value** | **Config** | **Value** |
| optimizer | AdamW | optimizer | AdamW |
| scheduler | CosLR | scheduler | CosLR |
| lr | 0.0005 | lr | 0.0005 |
| weight decay | 0.05 | weight decay | 0.05 |
| total epochs | 300 | total epochs | 350 |
| warmup epochs | 10 | warmup epochs | 10 |
| trans_dim | 384 | trans_dim | 384 |
| points | 1024 | points | 4096 |
| num_heads | 6 | num_heads | 6 |
| L token | 128 | L token | 128 |
| k | 4 | k | 4 |
| block | 4 | block | 4 |
| bs | 24 | bs | 16 |

## A2. Dataset Introduction

ModelNet40 (Wu et al., 2015) is a cornerstone benchmark for point cloud processing. It contains 12,000 3D models across 40 common object categories, such as airplanes and chairs, derived from CAD models and 3D scans. The dataset is formally divided into 9,800 models for training and 2,200 for testing. Its clean, standardized format has made it the de facto standard for training and evaluating 3D object classification algorithms.

ScanObjectNN (Uy et al., 2019) is a challenging benchmark for 3D object classification, built from real-world scanned scenes. Its approximately 15,000 objects (across 15 categories) are sourced from indoor and outdoor environments, resulting in point clouds with significant real-world complexities. These include variations in sampling density, high noise levels, and occlusions. Consequently, ScanObjectNN is primarily used to evaluate the robustness and generalization of classification algorithms under practical conditions, moving beyond the cleanliness of synthetic datasets.

ShapeNetPart (Yi et al., 2016) is a specialized subset of the ShapeNetCore dataset, designed for 3D object part segmentation. It comprises 16,881 models across 16 object categories, each annotated with detailed part-level labels. In addition to the geometric data, the dataset provides corresponding part annotations in JSON format. This high-quality, annotated data has established ShapeNetPart as a fundamental resource for training and evaluating networks for fine-grained part segmentation.

S3DIS (Armeni et al., 2016) is is a cornerstone benchmark for 3D indoor scene semantic segmentation. It consists of dense, large-scale LiDAR point clouds from 271 rooms across 6 building types, including offices and conference rooms. With over 2 billion points, each is annotated with one of 13 semantic categories (e.g., walls, floors, doors). This comprehensive and precise labeling provides critical support for developing and evaluating semantic segmentation algorithms in complex, real-world indoor environments.

## A3. Experiment Details

This study employs differentiated experimental configurations for point cloud classification and segmentation tasks, balancing task-specific requirements with experimental comparability. For **classification**, we use the AdamW optimizer with a Cosine Learning Rate scheduler, an initial learning rate of 0.0005, weight decay of 0.05, and train for 300 epochs (including 10 warm-up epochs). Each point cloud is sampled to 1024 points, with a batch size of 24. The model uses a token length L = 128 and comprises 4 blocks. For the **segmentation task**, core configurations—including the optimizer, learning rate, and weight decay—remain consistent with the classification setup. However, to meet the demands of detail preservation and convergence, we adjust the point cloud sampling to 4096 points, set the batch size to 16, and extend training to 350 epochs. For **few-shot classification** experiments on ModelNet40, we adopt the same configuration as Mamba3D to ensure a fair comparison. A detailed summary of the experimental configurations is provided in TableA1.

*Table A2.* Effect of $k$ in QELP on ModelNet40

| $k$ | OA (%) | mAcc (%) |
|---|---|---|
| 1 | 93.1 | 90.3 |
| 2 | 93.6 | 90.7 |
| 4 | **94.2** | **91.4** |
| 6 | 93.7 | 90.9 |

*Table A3.* Effect of Point Scale on ModelNet40

| $Scale$ | OA (%) | mAcc (%) |
|---|---|---|
| 512 | 92.6 | 89.6 |
| 1024 | **94.2** | **91.4** |
| 2048 | 93.8 | 90.8 |
| 4096 | 93.3 | 90.2 |

## A4. More Experiment

**Effect of Neighbor Scale.** We investigate the impact of the local neighborhood size $k$ in the QELP module through ablation studies on ModelNet40. As shown in Table A2, performance (OA and mAcc) improves as $k$ increases from 1 to 4, peaking at 94.2% OA and 91.4% mAcc. This indicates that a larger receptive field enhances feature representation. However, further increasing $k$ beyond 4 introduces redundant information, leading to performance degradation.

**Effect of Point Scale.** Currently, most deep learning point cloud methods employ a fixed sampling size of 1,024 points. To investigate the impact of this choice, we evaluated our network with 512, 1,024, 2,048, and 4,096 points. As shown in Table A3, performance peaks at 1,024 points (94.2% OA, 91.4% mAcc). Sampling with only 512 points provides insufficient surface coverage, limiting feature extraction, while larger sizes (2,048, 4,096) appear to introduce redundant information and computational burden, leading to a degradation in accuracy.

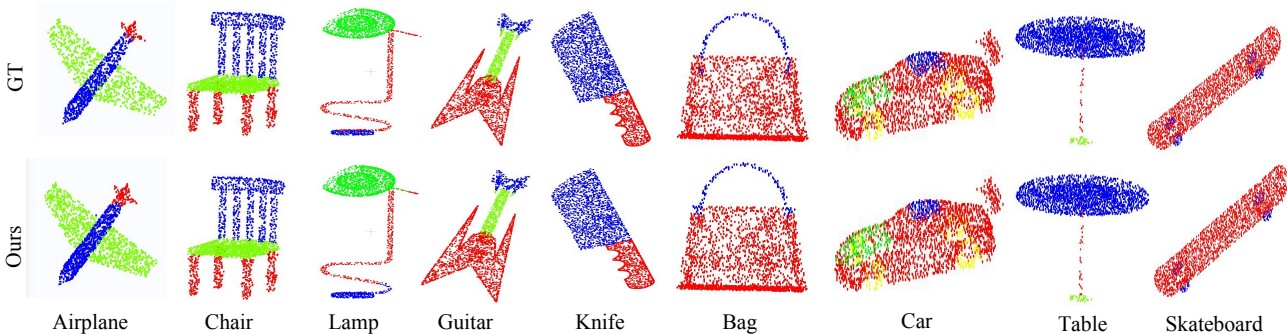

*Figure A1.* Part segmentation results on ShapeNetPart. Top row is ground truth and bottom row is our prediction.

## A5. More Visualization

We evaluate our method on ShapeNetPart, a benchmark for fine-grained, part-level segmentation. As visualized in Figure A1, our predictions (bottom row) align closely with the ground truth (top row) across diverse categories like Airplane, Chair, and Guitar. The model demonstrates precise boundary recognition and category distinction, accurately segmenting components such as airplane fuselages and wings, chair structures, and guitar parts. These results validate its strong capability in part-level feature extraction and segmentation.

