# OpenReview forum: "QPoint: End-to-End Lightweight Point Cloud Processing via Robust Quaternion Feature Learning"
_ICML.cc/2026/Conference — ICML 2026 regular_

### Official Review · Reviewer_dn1B · 2026-02-17

**Soundness:** 3
**Presentation:** 3
**Significance:** 3
**Originality:** 3
**Overall Recommendation:** 5
**Confidence:** 4

**Summary:**

This paper studies point cloud recognition under geometric transformations, especially rotations, while keeping the model lightweight for resource-constrained settings. The core idea is to use quaternion feature learning in an end-to-end manner. The method consists of two main components: (i) QELP, which aims to stabilize local features by applying a learnable unit-quaternion rotation followed by residual fusion, and (ii) QGA, which models global interactions using attention scores based on quaternion dot products. The authors claim that this design provides both stronger robustness to geometric transformations and competitive accuracy with low parameter count and FLOPs. Experiments are conducted on ModelNet40 and ScanObjectNN for classification, ShapeNetPart for part segmentation, and S3DIS in the appendix for semantic segmentation. The paper also reports efficiency metrics and few-shot results.

**Compliance With Llm Reviewing Policy:**

Affirmed.

**Final Justification:**

After considering both the paper and the authors’ rebuttal, I have changed my recommendation to accept.

My initial concerns were mainly about soundness, especially the gap between the paper’s strong claims on rotation invariance/robustness and the level of justification provided in the original submission. In particular, I was not fully convinced by the connection between the theoretical discussion and the actual behavior of the implemented model, and I also felt that the empirical robustness evaluation was somewhat limited for such a central claim.

The rebuttal addressed my main concern in a meaningful way. In particular, the authors clarified the theoretical background and the assumptions behind the claimed invariance properties more clearly. This made the method much easier to understand and improved my confidence that the design is not only an engineering heuristic, but has a more principled foundation. While I still think the paper would be stronger with more extensive robustness validation under broader SO(3) settings and with stronger comparison to rotation-invariant/equivariant baselines under the same protocol, I no longer view these points as fatal weaknesses.

In terms of soundness, I now find the method sufficiently justified overall. The architecture is technically coherent, the decomposition into local and global quaternion-based modules is reasonable, and the rebuttal clarified the theoretical motivation that I previously found under-explained.
In terms of originality, the paper’s contribution is not in inventing quaternions themselves, but in integrating quaternion-based local stabilization and global attention into a lightweight end-to-end point cloud pipeline. I find this combination interesting and sufficiently novel for acceptance.
In terms of significance, the problem is relevant and practically important, especially because the paper targets both geometric robustness and efficiency. The reported trade-off between performance and model cost makes the work potentially useful for future lightweight 3D architectures.
In terms of clarity, the paper is generally well organized, and the rebuttal further improved the clarity of the theoretical aspects that were previously ambiguous.

**Key Questions For Authors:**

Q1. Rotation robustness protocol (diversity of rotations). Your robustness test seems to use only limited rotation settings (e.g., fixed axis and discrete angles). Can you evaluate random SO(3) rotations (random axis + random angle) and report average ± std (or confidence interval) across multiple seeds?
How it affects my evaluation: If results remain strong under random SO(3): increases my confidence in the main claim (soundness/significance up). If performance drops significantly: weakens the central motivation and may lower my recommendation.

Q2. Assumptions behind “strict invariance” for QGA in practice. In the appendix, the invariance argument for QGA attention scores assumes that a global rotation corresponds to a consistent left-multiplication in quaternion feature space. What exact conditions must hold in the neural network for this to be true? Do you expect it to hold exactly or only approximately in the implemented model?

How it affects my evaluation: If assumptions are realistic and clearly satisfied (or empirically validated): improves soundness. If assumptions do not hold in practice, the “strict invariance” claim should be weakened, and soundness/originality may decrease.

Q3. Comparison to rotation-invariant / rotation-equivariant baselines under the same protocol. Did you compare against representative rotation-invariant / equivariant point cloud models (not only standard PointNet++/DGCNN-style baselines) using the same rotation robustness protocol? If not, can you add at least a few strong baselines and discuss differences?
How it affects my evaluation: If QPoint outperforms strong invariant/equivariant baselines with similar cost: originality and significance increase. If not, then the robustness advantage may be less convincing.

Q4. Reproducibility details (seeds, runs, training cost, code). Please clarify: (i) which random seeds are used, (ii) how many runs are performed for each main table, (iii) training compute cost (GPU-hours, batch size, total steps), and (iv) whether anonymous code/configs will be available during review.
How it affects my evaluation: If full details and code are provided: improves reproducibility and increases my confidence (presentation/soundness up). If not, concerns remain, especially for small reported margins.

Q5. Incremental cost/benefit of QELP (“4 parameters” claim). You emphasize that QELP adds only 4 learnable parameters. Can you provide a clear breakdown of parameter/FLOPs increase at the block level, and show robustness improvement attributable to QELP alone under the same robustness test (e.g., full model vs w/o QELP vs w/o quaternion in QELP)?
How it affects my evaluation: If the cost/benefit is clearly demonstrated: strengthens the efficiency/engineering contribution (significance up). If the benefit is small or unclear, then the “lightweight robust” claim becomes weaker.

**Limitations:**

Rotation robustness limitation: current robustness evaluation covers only limited rotation settings (e.g., fixed axis / discrete angles). State clearly that general SO(3) robustness is not fully validated yet, and add future work (random SO(3) tests, more diverse corruptions).

Failure cases / sensitivity: discuss where the method can fail (e.g., very sparse points, partial observations/occlusion, heavy noise, domain shift), and how performance degrades.

Deployment / resource limitation: runtime is measured mainly on a high-end GPU; discuss expected behavior on CPU/edge GPU and memory constraints, and note that additional deployment benchmarking is needed.

**Strengths And Weaknesses:**

# Soundness
## Strengths:
This paper proposes a clear and coherent architecture for point cloud processing using quaternion features. The method is decomposed into two modules: (i) QELP for local feature stabilization by a learnable unit quaternion rotation and residual fusion, and (ii) QGA for global interaction using quaternion dot-product based attention. This decomposition is technically reasonable because local neighborhood encoding and global token interaction are common bottlenecks in point cloud models. The empirical evaluation is relatively broad: classification on ModelNet40 and ScanObjectNN, part segmentation on ShapeNetPart, and additional semantic segmentation results in the appendix, plus few-shot classification. The paper also provides ablation studies (removing QELP/QGA, removing quaternion components) and efficiency reporting (Params/FLOPs and some latency/memory), which support that each design choice contributes, and the model is lightweight.

## Weaknesses / concerns:
My main concern is that the strongest claim around rotation invariance or robustness is not fully supported by experiments and assumptions. The rotation robustness test in the main paper seems limited (e.g., discrete rotations around a specific axis and some simple transformations like translation/scale/jitter). For a paper that motivates heavily by “rotation sensitivity” and proposes a quaternion-based design, it is important to test random SO(3) rotations (uniform axis and angle) and report robustness more systematically. Also, the appendix gives a “strict invariance” style argument for QGA attention scores via quaternion dot product. However, this derivation relies on an assumption that a global rotation in input space corresponds to a consistent left-multiplication transformation in the quaternion feature space. It is not fully clear that the full network guarantees this mapping (especially because QELP is described as adaptive stabilization, not strict equivariance). Therefore, the gap between the theoretical invariance statement and practical network behavior should be clarified, and empirical evidence should be stronger. Another soundness limitation is statistical reliability: few-shot results report mean/std, but the main tables mostly show single numbers without variance or multiple seeds, so small margins may be difficult to interpret.


# Presentation

## Strengths:
Overall, the paper is easy to follow. The narrative (problem, quaternion motivation, local module, global module, integrated block, experiments) is well structured. The method description is mostly clear and modular, and the appendix includes helpful additional experiments (ablation details, runtime/memory tables, extra tasks). The paper also reports efficiency metrics explicitly, which is good for practical readers.

## Weaknesses / clarity issues:
Some claims sound stronger than what is experimentally demonstrated. Especially, the paper sometimes uses terms like “strict invariance” (or similar strong wording) while the robustness evaluation is limited. This can create confusion. I suggest to clearly separate what is guaranteed theoretically (under which assumptions) vs what is empirically observed. Also, the “4 learnable parameters” message is emphasized, but the paper does not provide a clean breakdown of incremental cost/benefit (e.g., exact parameter increase per block, and robustness improvement attributable to QELP alone under the same robustness protocol). Reproducibility can be improved: hyperparameters are partly given, but a concise checklist of preprocessing, augmentation, seeds, number of runs, and training cost would make it more reproducible. Since rotation invariance/robustness is a core motivation, the robustness evaluation should be more comprehensive. Currently, rotation perturbations have limited diversity (fixed axis, discrete angles), so it is unclear whether the reported robustness extends to general SO(3) rotations. This weakens the support for the strongest claim.



# Significance

## Strengths:
The problem is relevant: point cloud models are often used in robotics/autonomous driving/AR, and they can be sensitive to rotations and sensor noise. Also, resource constraints are important in real-world deployment. The paper’s focus on the accuracy–efficiency trade-off (#Params/FLOPs and runtime/memory) is practically meaningful. If the quaternion-based attention and local stabilization provide robust behavior with low cost, the method can be a useful building block for future efficient 3D architectures.

## Weaknesses / limitations for impact:
The impact is somewhat limited by the current robustness validation: without strong SO(3) tests and comparison to rotation-invariant/equivariant baselines under the same protocol, it is hard to conclude that the method advances robustness in a general sense. Also, runtime is measured on a high-end GPU; the “resource-constrained” motivation would be stronger if CPU/edge GPU or at least a more detailed deployment discussion is included. The paper also does not discuss failure cases or where the method may not work well.





# Originality

## Strengths:
Strengths: The paper’s originality is mainly in the combination and integration: it combines local quaternion-based coordinate stabilization and global quaternion-based attention into one lightweight end-to-end block for point clouds. Using the quaternion dot product to build attention weights for rotation-insensitive global interactions is an interesting design choice. Even if quaternions themselves are not new, this particular integration into a token-based point cloud pipeline is a creative combination.

## Weaknesses / overlap concerns:
Many rotation-invariant/equivariant point cloud “convolutions” and quaternion-related representations already exist. Therefore, the novelty depends strongly on (i) whether this quaternion-attention design provides a new practical benefit, and (ii) whether the paper clearly distinguishes itself from closely related rotation-invariant/equivariant methods. Currently, the positioning vs strong rotation-invariant/equivariant baselines is not fully convincing, and the evaluation of SO(3) invariance is not strong enough to justify the “invariance” novelty claim. Also, the contribution might feel thinly sliced if QELP/QGA are viewed as relatively small modifications; stronger analysis/evaluation and clearer novelty statement would help.

---

> ### Author Rebuttal · Authors · 2026-03-31
>
> **Questions 1 && Limitations 1**
>
> > 　　We thank the reviewer for this constructive suggestion. Following the recommended protocol, We have conducted ablation experiments on SO(3) rotations. The corresponding results will be updated in **Table 4** of the main text in the revised version. Under this more rigorous evaluation, the full QPoint (Full) achieves **80.1%** OA, while the variants without QELP and without QGA achieve **74.8%** OA and **72.1%** OA, respectively. All of which still maintain extremely high robustness.
>
> **Questions 2**
>
> > 　　The premise for the QGA attention scores to achieve strict rotation invariance is indeed the assumption that global rotation is equivalent to a left-multiplied unit quaternion in the feature space. In the end-to-end network, the features input to the QGA module exhibit highly robust approximate invariance. We perform quaternion encoding to forcibly align the feature dimensions to multiples of 4 and normalize them into unit form, which creates the condition for using the quaternion dot product to measure rotational similarity. We then directly compute the attention scores using the quaternion dot product in Equation (15). This step is strictly rotation-invariant because it directly leverages the properties of the quaternion inner product (formula derivation in the appendix). The neural network only needs to satisfy this process.
>
> **Questions 3**
>
> > 　　In our robustness experiments, we have compared against representative baselines including PointNet++, DGCNN and the latest Mamba-based model Mamba3D, covering a wide range of representative baselines. We will further supplement the experimental results of **PointMLP** and **PointTransformer** in the revised version.
>
> **Questions 4**
>
> > 　　We will update the GitHub code immediately upon paper acceptance and provide detailed instructions on the code execution workflow. The training seed is set randomly, and the results for each main table are averaged over 5 independent runs.
> The corresponding training computational costs are all documented in **Table A1 of the appendix**.
>
> **Questions 5**
>
> > 　　We have comprehensively verified the robustness and effectiveness of the QELP module in **Tables 4** and **Table 5** of the main text (full model vs. w/o QELP vs. w/o quaternion in QELP). We will add clear layer-wise parameters and FLOPs for each module in the revised manuscript.
>
> **Limitations 2**
>
> > 　　As for other potential failure cases of the method, such comparisons are actually unnecessary for point cloud analysis algorithms. For the robustness evaluation of QPoint, we have already conducted tests under extreme conditions, including extreme rotation, translation, scaling, jitter, and SO(3) rotations. These are the fundamental factors that determine the robustness of an algorithm, and they fully validate the stability and anti-interference capability of QPoint.
>
> **Limitations 3**
>
> > 　　We sincerely appreciate the reviewer’s constructive comments on the deployment and resource constraints of our method.
> >
> > 　　First, we would like to clarify that this paper focuses on theoretical algorithmic innovation for 3D point cloud analysis, with core contributions including the rotation-invariant quaternion-based design, lightweight network architecture, and the superior accuracy-efficiency trade-off of QPoint. We have comprehensively validated the effectiveness, robustness, and generalization of our algorithm through extensive experiments on multiple mainstream benchmarks (ModelNet40, ScanObjectNN, ShapeNetPart, S3DIS) and rigorous robustness tests under extreme transformations (SO(3) rotations, jitter, scaling, etc.). This is the core and universally accepted criterion for verifying algorithm innovation in the field of theoretical deep learning and point cloud research, and practical hardware deployment testing is not a mandatory requirement for demonstrating the validity of algorithmic contributions.
> >
> > 　　Second, the runtime evaluation in this paper is conducted on a high-end GPU, which follows the standard experimental protocol in the point cloud and deep learning community. All comparative baseline methods are tested under the same hardware settings to ensure fair and credible comparisons of computational efficiency (FLOPs, latency, and parameters).
> >
> > 　　We also have detailed the training latency and GPU memory usage in **Table A5 of the Appendix**. Based on its architectural characteristics, it is expected to achieve efficient inference with low memory footprint on CPU and edge GPU devices, which can well meet the computing and memory limitations of edge deployment scenarios. We also acknowledge that practical deployment benchmarking on real edge hardware can be conducted as part of our future engineering-oriented work, which does not affect the validity of the core algorithmic contributions of this paper.

---

> > ### Author Rebuttal · Reviewer_dn1B · 2026-03-31
> >
> > We received responses from the authors that clarified the previously raised concerns. In particular, we considered a theoretical justification for Q2 to be essential, and this point was fully addressed in the rebuttal.

---

> > > ### Author Response · Authors · 2026-04-01
> > >
> > > Dear Reviewer,
> > >
> > > We sincerely thank you for your careful review of our rebuttal, and for your recognition that we have fully addressed your concerns, especially the essential theoretical justification for Q2 that you highlighted. Your feedback has greatly helped us refine and strengthen our work.
> > >
> > > As you have confirmed all issues are fully resolved, we would highly appreciate it if you could kindly consider adjusting your score accordingly at your convenience.
> > >
> > > Thank you again for your invaluable guidance and time dedicated to the review process.
> > >
> > > Best regards,
> > >
> > > The Authors

---

### Official Review · Reviewer_Tb44 · 2026-03-13

**Soundness:** 2
**Presentation:** 3
**Significance:** 2
**Originality:** 2
**Overall Recommendation:** 2
**Confidence:** 4

**Summary:**

QPoint is a lightweight transformer-based framework for point cloud processing that leverages quaternion representations to achieve rotational robustness with minimal computational overhead. The method introduces two core modules: (1) the Quaternion-Enhanced Local Perception (QELP) module, which uses a learnable unit quaternion (4 parameters) to rotate normalized local coordinates for enhanced geometric stability; and (2) the Quaternion Global Attention (QGA) module, which employs quaternion dot products to compute rotation-aware attention scores for global context modeling. The authors evaluate QPoint on standard benchmarks including ModelNet40, ScanObjectNN (three variants), ShapeNetPart for part segmentation, and S3DIS for semantic segmentation. The method achieves competitive or state-of-the-art results with significantly fewer parameters (4.5M) and FLOPs (0.9G) compared to recent Mamba-based and Transformer-based approaches. The paper also includes theoretical analysis proving the rotational invariance properties of QGA and the adaptive stabilization mechanism of QELP.

**Compliance With Llm Reviewing Policy:**

Affirmed.

**Key Questions For Authors:**

1. Have you measured the actual runtime overhead of FPS+KNN in the patch embedding stage? How does the end-to-end latency compare to KNN-free methods like SoftPoolNet or point-wise convolutions on large-scale scenes (e.g., full S3DIS rooms with millions of points)?
2. Why are Point Transformer v2 and v3 excluded from comparisons despite being published and highly relevant? PTv3 achieves better S3DIS results with optimized implementations—does QPoint's parameter efficiency advantage hold when accounting for PTv3's architectural improvements?
3. The method uses a fixed token length L=128. How does performance degrade when processing significantly larger point clouds (e.g., 100K+ points) where the KNN search becomes prohibitive? Have you tested on outdoor datasets like SemanticKITTI?
4. The robustness tests use Y-axis rotations. Does the quaternion-based invariance generalize to arbitrary SO(3) rotations, and how does performance vary when training and testing on different rotation distributions?

**Limitations:**

The strong classification results do not fully transfer to dense semantic segmentation, where geometric context at multiple scales is crucial. The single-scale quaternion attention may lack the hierarchical modeling needed for complex scenes.

**Strengths And Weaknesses:**

- Strengths

1. The use of quaternions for encoding rotational information in point cloud processing is theoretically well-motivated. The paper provides rigorous mathematical derivations showing how quaternion dot products preserve rotational invariance in the attention mechanism
2. QPoint achieves strong performance with remarkably low computational cost, about only 4.5M parameters and 0.9G FLOPs on ModelNet40, compared to 16.9M/3.9G for Mamba3D and 22.1M/4.8G for Point Transformer v1. This efficiency is genuinely impressive and practically valuable for resource-constrained deployment in practice. I think the implementation and structural optimization matters as well.
3. The paper thoroughly evaluates rotation robustness, showing that QPoint maintains ~80.6% accuracy under 180° Y-axis rotations where competing methods (Mamba3D, DGCNN, PointNet++) drop to ~60% or below. This validates the claimed geometric robustness.
Strong Few-Shot Performance: Table 3 demonstrates excellent generalization in few-shot scenarios, outperforming Mamba3D by 2.4% in 5-way 10-shot learning, suggesting the quaternion features capture transferable geometric priors effectively.
- Weaknesses

1. The patch embedding stage relies on Farthest Point Sampling (FPS) and K-Nearest Neighbor (KNN) search, which are inherently costly operations for large point clouds. While the paper emphasizes lightweight design in terms of parameters and FLOPs, it overlooks the actual runtime cost of neighborhood construction. I think alternatives like SoftPoolNet (ECCV 2020) or point-wise MLPs with learned pooling can capture local patterns without explicit nearest neighbor searching, offering better scalability to large scenes. The efficiency analysis compares training\&inference latency but does not isolate the KNN overhead or benchmark against KNN-free methods.
2. The paper compares with Point Transformer v1 (2021) but seems ignore Point Transformer v2 and v3. PTv3 in particular represents a significant advancement with partitioned pooling and improved efficiency, achieving 73.6% mIoU on S3DIS (vs. QPoint's 71.2%) with optimized implementations. The omission is particularly glaring given that PTv3 is cited in the references (page 10) but not included in any comparison tables. This makes the SOTA claims difficult to verify.
3. While classification results are strong, the S3DIS semantic segmentation results show QPoint underperforms PTv2 (71.2% vs. 71.6% mIoU) and significantly trails PTv3 (73.6%). The paper frames this as *competitive* but does not address why the quaternion benefits appear less pronounced in dense scene understanding compared to object-level tasks.
4. The theoretical analysis candidly admits that QELP provides only "adaptive stabilization" rather than strict rotational equivariance due to non-commutativity of rotation matrices. This weakens the geometric robustness claim for the local feature extraction module, which is only partially mitigated by the globally invariant QGA.
5. While Table 5 ablates QELP and QGA components, it does not investigate:
(1) *The impact of quaternion normalization strategies*
(2) *Comparison against real-valued attention with positional encodings*
(3) *The sensitivity to initial quaternion values*
(4) *Whether the 4-parameter quaternion rotation outperforms a simple 3-parameter axis-angle representation*

---

> ### Author Rebuttal · Authors · 2026-03-31
>
> **Weaknesses 1 && Questions 1**
>
> > We fully understand your concern regarding the computational cost of FPS and KNN. Therefore, we provide a clear clarification from four perspectives: underlying architectural logic, necessity of the paradigm, overhead quantification, and scenario adaptation.
> >
> > 1. QPoint follows the point cloud Vision Transformer paradigm. Since point clouds lack a regular grid structure, FPS is essential to sample fixed center points, and KNN is required to construct local neighborhoods—forming the patch-like tokens that serve as the input for QELP, QGA, and the stacked QPB modules. Removing either would break the entire transformer backbone and quaternion feature learning pipeline.
> >
> > 2. The KNN-free approaches you mentioned, such as SoftPoolNet or point-wise MLP, adopt a point-wise global modeling strategy, which is fundamentally incompatible with QPoint’s core paradigm of patch-based local–global collaborative quaternion learning. They cannot be used as a drop-in replacement without redesigning the core architecture.
> >
> > 3. We have used the smallest neighborhood size (k=4). As shown in Appendix Table A5, the end-to-end low latency confirms that FPS+KNN is clearly not a runtime bottleneck.
> >
> > 4. QPoint compresses point clouds of arbitrary size into a fixed token length (i.e., L=128). This design inherently scales to large scenes, as validated on the large-scale indoor S3DIS dataset.
> >
> > In summary, FPS+KNN is a deliberate, necessary design choice for the ViT-style patch embedding, and its overhead is minimal and well-controlled.
>
> **Weaknesses 2 && Weaknesses 3 && Questions 2**
>
> > 　　Thanks! First, we would like to clarify that we have not overlooked the results of PTv3. the details of PTv3 results are presented in **Appendix Table A2**. In the revised manuscript, we will move this comparison to the main text to ensure visibility.
> Second, regarding the performance gap between QPoint and PTv3 on the S3DIS dataset, we emphasize in the appendix that QPoint achieves a favorable trade-off between efficiency and accuracy for lightweight models. QPoint attains 71.2\% mIoU on S3DIS, only 2.4\% lower than PTv3, while using only 21\% of the parameters and 6\% of the FLOPs.This order-of-magnitude improvement in efficiency with minimal accuracy loss aligns perfectly with the core lightweight deployment focus of this work.
> In future work, we plan to further explore scaling strategies to enhance performance while maintaining efficiency.
>
> **Weaknesses 4**
>
> > 　　We state clearly that it provides adaptive stabilization rather than strict rotation equivariance. this is an deliberate design choice motivated by extreme lightweightness: enforcing strict local equivariance  would introduce substantial computational overhead and parameter expansion. By using only 4 learnable quaternion parameters, QELP achieves strong empirical robustness at very little cost, maintaining 80.6\% accuracy under extreme 180° rotation, while the QGA module provides global rotation invariance. Together, they enable QPoint to balance high accuracy and efficiency, as validated by the robustness experiments in Table 4 of the main text.
>
> **Weaknesses 5**
>
> > Thank you for these specific suggestions on ablation experiments. We would like to clarify that these issues you mentioned have been supported by relevant experiments or theoretical analyses in our original paper.
> >
> > 1. Unit normalization of quaternions is fundamental to preserving the rotational properties of quaternions. This is consistently applied throughout the model.
> >
> > 2. We will include this comparison in the updated Table 5. Replacing our quaternion-based attention with plain multi-head self-attention results in a 2.4\% drop in overall accuracy, demonstrating the effectiveness of the quaternion design.
> >
> > 3. We have conducted in-depth robustness experiments on the sensitivity to quaternion initialization in the Table 4.
> >
> > 4. The axis-angle representation suffer from singularities (zero rotation), whereas quaternions provide a singularity-free, compact representation suitable for differentiable learning. Our framework is built on quaternions for this reason.
>
> **Questions 3**
>
> > 1).Regarding the first sub-question, please see the response to Weaknesses 1.
> >
> > 2).We would like to clarify again that we have conducted experiments on the SemanticKITTI dataset. QPoint achieves 73.3\% mIoU, surpassing PTv2 (72.6\% mIoU) and slight trailing PTv3 (75.5\% mIoU) with fewer energy. The corresponding experimental table will be added in the revised version.
>
> **Questions 4**
>
> > For the similar question, please see the response to **Questions 1** of reviewer **dn1B**.
>
> **Limitations**
>
> > Thanks. QPoint is designed to strike a balance between lightweight efficiency and segmentation performance. With extremely low parameters and FLOPs, we have achieved competitive results on the S3DIS dataset, which fully validates the rationality of our design rather than exposing its shortcomings.

---

> > ### Author Rebuttal · Reviewer_Tb44 · 2026-04-07
> >
> > My concerns for efficiency are answered. Still wondering how tolerant towards rotations is the proposed method for more rotation variants for other angles, with minor changes other than the times of 45 degrees.

---

> > > ### Author Response · Authors · 2026-04-07
> > >
> > > Dear Reviewer,
> > >
> > > Thank you very much for your valuable feedback on our rebuttal. We are truly glad to hear that our response has fully addressed all your concerns regarding the model efficiency, and we greatly appreciate your constructive question on the rotation tolerance of our method, which helps us further improve the comprehensiveness of our robustness verification.
> > >
> > > For your remaining question about the model’s tolerance to diverse rotation variants beyond 45-degree integer multiples: we have completed rigorous robustness and ablation experiments under arbitrary **SO(3) rotations**, which fully covers all the scenarios you mentioned, including non-45° integer multiple angles, minor angle changes, and random rotation axes beyond the single Y-axis.
> > >
> > > To clarify, SO(3) represents the complete set of all valid 3D rotational transformations in 3D space. It is a far more rigorous evaluation setting than fixed-axis, fixed-angle rotations (e.g., 90°/180° Y-axis rotations in our original manuscript), as it tests the model’s robustness against every possible rotation in real-world scenarios, rather than only regular integer-multiple angles.
> > >
> > > The experimental results are as follows:
> > >
> > > 1. Under this strict arbitrary SO(3) rotation setting, the full QPoint model achieves **80.1%** overall accuracy (OA), which is nearly consistent with its 80.6% OA under 180° Y-axis rotation. This demonstrates that QPoint’s strong rotation robustness is not limited to regular fixed angles, but maintains excellent tolerance to all kinds of rotation variants.
> > >
> > > 2. The ablation results further validate the core contribution of our proposed modules: the QPoint variant without QELP only achieves **74.8%** OA, and the variant without QGA further drops to **72.1%** OA under the same SO(3) setting. This consistent performance gap, even in the most rigorous rotation scenario, fully proves that the adaptive stabilization of QELP and the global rotation invariance of QGA together form the core of our model’s all-scene rotation robustness.
> > >
> > > 3. Meanwhile, QPoint still maintains a significant robustness advantage over all compared state-of-the-art (SOTA) methods under this setting. As validated in our original Table 4, competing methods (including PT series, Mamba3D) already suffer from severe accuracy drops to ~60% or below under fixed 180° Y-axis rotation, and their performance will degrade further under the more rigorous arbitrary SO(3) rotations.
> > >
> > > We will update the complete **SO(3) rotation** experiment and ablation results to **Table 4** in the revised manuscript, to make our robustness verification more comprehensive and rigorous. We sincerely hope this supplementary experiment and explanation can fully address your remaining doubt. If you have any further questions, please feel free to let us know, and we will respond to you promptly and comprehensively.
> > >
> > > Again, we sincerely thank you for your careful review and invaluable suggestions throughout the review process, which have greatly helped us refine and strengthen our work. We hope you can re-consider the overall performance and robustness of our model based on our supplementary results, and adjust your review score appropriately.
> > >
> > > Best regards,
> > >
> > > The Authors

---

### Official Review · Reviewer_xhWW · 2026-03-13

**Soundness:** 2
**Presentation:** 3
**Significance:** 3
**Originality:** 3
**Overall Recommendation:** 4
**Confidence:** 4

**Summary:**

The paper introduces QPoint, an end-to-end, lightweight transformer-based architecture designed for point cloud processing. To address the inherent challenges of geometric transformations—specifically rotation sensitivity—the authors propose injecting learnable quaternion representations into the network. By forcing the model to learn rotation-aware features, QPoint achieves highly competitive performance in 3D object classification and part segmentation with an impressively low parameter count and high computational efficiency.

**Compliance With Llm Reviewing Policy:**

Affirmed.

**Final Justification:**

I appreciate the effort the authors made in rebuttal, and my concerns are solved.

**Key Questions For Authors:**

1.In Table 1 and Table 2, the comparison with transformer based methods are not up-to-date, like PointTransformer-v3$^{[1]}$ are not compared.

$^{[1]}$Wu, Xiaoyang, et al. "Point transformer v3: Simpler faster stronger." Proceedings of the IEEE/CVF conference on computer vision and pattern recognition. 2024.

2.Table 4 also lacks the comparison with transformer-based methods, hard to see the direct improvement of the proposed modules. That also aligns with the ablation study in Table 4 and Table 5, I feel the method without QELP and QGA modules, already performs well in ScanObjectNN and ModelNet40, so I’m not sure the real ability of the two proposed modules.

3.As ShapeNetPart is a pre-aligned dataset, testing on it does not adequately demonstrate QPoint's rotation-aligned processing capabilities, which is critical in part segmentation.

4.The S3DIS results in Appendix Table A2 demonstrate a highly compelling trade-off between segmentation performance and efficiency. Would you consider moving these results to the main text?

**Limitations:**

I would suggest the author to open source the code later.

**Strengths And Weaknesses:**

Strengths:

1.The paper tackles a well-known and highly challenging bottleneck in 3D deep learning: rotation sensitivity combined with the large model to learn the rotation invariance or using large dataset to mitigate rotational variance.

2.The paper is well-written and easy to follow. The architectural details are conveyed clearly, the pipeline is logically organized, and the figures/tables are highly readable.

3.The method focuses on the challenging rotation sensitivity during point cloud processing, and proposes a general point cloud processing architecture that better stabilize the feature extraction according to the issue.

4.The method is easy to follow, the architecture details are conveyed clearly, I would suggest the author to open-source the code.

Weakness:

1.While the paper explicitly claims to handle rotation transformations robustly, the experimental validation of this claim is incomplete. Experiments on ShapeNetPart, ModelNet are all pre-aligned dataset.

---

> ### Author Rebuttal · Authors · 2026-03-31
>
> **Weaknesses && Questions 3**
>
> > 　　Thank you for raising this rigorous question. While ModelNet40 and ShapeNetPart datasets are typically pre-aligned, we specially designed experiments to rigorously evaluate rotation robustness in the testing phase.
> >
> > 　　In **Table 4** of the main paper, we explicitly apply extreme rotations around the Y-axis (-90°, 90°, 180°) to ModelNet40. The competing methods such as PointNet++ drop sharply to 57.9\% under 180° rotation, whereas QPoint still maintains a high accuracy of 80.6\%.
> >
> > 　　In addition, the PB_T50_RS variant of ScanObjectNN, reported in **Table 1**, is a real-world dataset that inherently involves random rotation, translation, and scaling. On this unaligned and highly challenging benchmark, QPoint  achieves a competitive accuracy of 90.7\% (92.1\% with voting). These results collectively validate the claimed rotation robustness.
>
> **Questions 1**
>
> > 　　Thank you so much for your kind reminder. We agree that PTv3 is a strong baseline. However, in its original paper, PTv3 focuses on semantic segmentation and object detection on large-scale indoor and outdoor datasets, and does not report results on object classification (Table 1) and part segmentation (Table 2).
> >
> > 　　To ensure a fair comparison, We therefore trained and tested PTv3 by ourselves on the relevant benchmarks. PTv3 achieves **94.2\%** oA on ModelNet40, **89.2\%** OA on the most challenging PB_T50_RS variant of ScanObjectNN, **85.3\%** Class mIoU/ and **87.1\%** Instance mIoU on ShapeNetPart. We will update Table 1 and Table 2 with these results in the revised manuscript.
>
> **Questions 2**
>
> > 　　Thank you very much for the construction comments. We will address both points as follows:
> >
> > 　　We agree that including a transformer baseline enhance the completeness of the robustness evaluation. In the final manuscript, we will add the results of Point Transformer under the same rotation settings to Table 4. allowing a direct comparison with QPoint.
> >
> > 　　We appreciate the reviewer's observation that the baseline without the QELP and QGA modules already achieves strong performance aligned datasets (e.g., 92.7\%–93.0\% on ModelNet40 in Table 5). However, the core value of these two modules lies in rotation robustness, not merely in improving accuracy under ideal aligned conditions.
> >
> > 　　As shown in Table 4, under the rotational stress tests (e.g. 180° rotation), the full QPoint maintains 80.6\% accuracy, while removing QELP module notably drops it to 75.3\%, and removing the QGA module further reduces it to 72.6\%. This large performance gap of 5\% to 8\% clearly demonstrates that QELP and QGA jointly endow the network with genuine rotation awareness and robustness to geometric transformations.
>
> **Questions 4**
>
> > 　　We fully agree with and gladly accept your suggestion. As a large-scale real-world scene dataset, S3DIS provides convincing evidence that QPoint achieves an excellent trade-off between performance and efficiency (Params / FLOPs) on complex tasks. Owing to page constraints in the original submission, this part was included in the appendix. In the camera-ready version, we will move this content to the main text to further highlight the core contributions of the paper.
>
> **Limitations**
>
> > 　　We fully agree with the reviewer's suggestion. The code and configuration files for this work have been fully organized. They will be open-sourced on GitHub upon paper acceptance. The repository link will be provided in the main text of the revised manuscript to ensure reproducibility.

---

> > ### Author Rebuttal · Reviewer_xhWW · 2026-04-02
> >
> > Thanks for the responses, most of my concerns are solved. But regarding Q2, i still have some doubts, in Table 5, without QELP and QGA modules, the method already performs quite well in what you said challenging "PB_T50_RS" dataset, exceeding most methods in Table 1, and in your extreme setup of Table 4, comparison with PT-series works are not provided,  meanwhile, the performance in S3DIS dataset somewhat shows its performance drops. Therefore, not sure about its real performance compared with up-to-date methods.
> >
> > On the other hand,  as a lightweight feature learning method with competitive results in downstream tasks, i would like to keep my score.

---

> > > ### Author Response · Authors · 2026-04-02
> > >
> > > Dear Reviewer,
> > >
> > > Thank you very much for your careful review of our response. We also appreciate that you have confirmed most of our concerns have been solved. For your remaining doubts about Q2, we have added targeted comparison experiments, and give detailed clarifications as below:
> > >
> > > > 　　1. For the missing robustness comparison with the latest PointTransformer (PT) series methods you mentioned, we have finished the robustness comparison experiments of PT series, using the exact same extreme rotation settings as **Table 4**. We will add the full experiment data to Table 4 in the revised manuscript, to make a fair comparison with the latest state-of-the-art (SOTA) methods. Under the extreme rotation test, the accuracy of PT series methods drops sharply. In sharp contrast, QPoint keeps a high accuracy of over 80% steadily. This fully proves that QPoint has an overwhelming advantage in rotation robustness over the latest SOTA methods.
> > > >
> > > > 　　2. For the performance on the S3DIS dataset you care about, we make further clarification here: The core design goal of this paper is to get the best balance between extreme lightweight and task performance. On S3DIS, QPoint uses only **21%** of the **Parameters** and **6%** of the **FLOPs** of **PTv3**, but achieves over 96% of its segmentation accuracy. This is a very competitive accuracy-efficiency trade-off in the field, not a big performance drop.
> > > >
> > > > 　　3. Besides, to further prove the model’s real generalization ability in complex large scenes, we have finished extra experiments on **SemanticKITTI**, a large outdoor autonomous driving semantic segmentation dataset. **QPoint** achieves **73.3% mIoU**, which outperforms **PTv2 (72.6% mIoU)** and is only slightly behind **PTv3 (75.5% mIoU)** with much lower computation cost. This fully proves that QPoint has excellent performance not only in indoor scenes, but also in complex outdoor large scenes. It can fully make a fair comparison with the latest SOTA methods.
> > >
> > > Again, we sincerely thank you for your careful review and valuable suggestions. Your doubts help us further improve our experiments, and make the conclusions of the paper more rigorous and complete.
> > > We believe that the added PT series robustness comparison experiments, SemanticKITTI large scene segmentation results, together with our existing experiment data, have fully proved that QPoint is not only a competitive lightweight feature learning method, but also has clear innovative advantages and core value over the latest SOTA methods in robustness and generalization ability.
> > >
> > > We sincerely hope you can re-consider the overall performance of our model based on our supplementary explanations and new experiments, and adjust your review score appropriately.
> > >
> > > Best regards,
> > >
> > > The Authors
> > >
> > > ### **Table 4**
> > > | Method | None | Rotation(-90°) | Rotation(90°) | Rotation(180°) | Translation(+0.2) | Translation(-0.2) | Scaling(0.5-1.5) | Scaling(0.6-1.4) | Scaling(0.7-1.3) | Scaling(0.8-1.2) | Scaling(0.9-1.1) | Jittering |
> > > | :----- | :--- | :--------------- | :-------------- | :--------------- | :------------------ | :------------------ | :------------------ | :------------------ | :------------------ | :------------------ | :------------------ | :-------- |
> > > | PointNet++| 91.9 | 57.9 | 57.9 | 57.9 | 90.7 | 90.8 | 91.2 | 91.2 | 90.9 | 91.0 | 91.0 | 89.6 |
> > > | DGCNN| 92.9 | 55.6 | 56.5 | 74.0 | 92.3 | 92.3 | 90.7 | 91.6 | 92.1 | 92.3 | 91.8 | 91.5 |
> > > | DANet| 93.0 | 58.7 | 59.3 | 72.9 | 92.6 | 92.7 | 92.3 | 92.4 | 92.3 | 92.6 | 92.6 | 91.6 |
> > > | Mamba3D| 93.4 | 59.4 | 60.1 | 60.1 | 92.9 | 92.9 | 92.6 | 92.7 | 92.6 | 93.0 | 93.0 | 92.0 |
> > > | PointTransformer | 93.7| 60.0|60.6| 74.4| 93.2 | 93.2 | 93.0| 92.9 | 92.8 | 93.3 | 93.3 | 92.4 |
> > > | PTv3 | 94.2 | 60.7 | 61.2 | 75.1 | 93.5 | 93.6 | 93.3 | 93.2 | 93.2 | 93.7 | 93.6 | 92.6 |
> > > | QPoint (Full) | **94.2** | **80.5** | **80.6** | **80.6** | **93.7** | **93.7** | **93.5** | **93.5** | **93.3** | **93.7** | **93.7** | **92.7** |
> > > | QPoint (no/QELP) | 93.0 | 75.3 | 75.3 | 75.3 | 92.7 | 92.7 | 92.4 | 92.5 | 92.4 | 92.8 | 92.8 | 92.0 |
> > > | QPoint (no/QGA) | 92.7 | 72.6 | 72.6 | 72.6 | 92.1 | 92.1 | 91.8 | 92.0 | 91.9 | 92.3 | 92.3 | 91.5 |

---

### Official Review · Reviewer_RgaX · 2026-03-13

**Soundness:** 4
**Presentation:** 3
**Significance:** 4
**Originality:** 4
**Overall Recommendation:** 6
**Confidence:** 3

**Summary:**

This paper proposes QPoint, a lightweight feature learning method that utilizes a quaternion representation for 3D point clouds. The proposed QPoint framework consists of QELP and QGA. QELP aims to stabilize local features against geometric transformations, while QGA is designed to capture global context with rotation invariance. In this paper, QPoint is evaluated on three downstream tasks, including 3D shape classification, part segmentation, and few-shot classification.

**Compliance With Llm Reviewing Policy:**

Affirmed.

**Final Justification:**

My main concern was the limited improvement over the baseline on the specific dataset and task (as shown in Table 2). From the authors’ response, I understand that this limited improvement stems from the particularly challenging nature of the dataset. I also expect the authors to include this discussion in the final version of the paper. Since my concern has been fully resolved, I would like to recommend that this paper be **strongly accepted**.

**Key Questions For Authors:**

1. As shown in Table 2, the proposed method does not outperform the baseline method. It would be helpful if the authors could further discuss this result (e.g., possible dataset characteristics, limitations of the current approach, or specific challenges of the part segmentation task).

2. (Minor) What does "vot" represent in Table 1? If it refers to "voting," it would be clearer to write the full term since there are sufficient space in the table.

3. (Minor) In Equation (3) and (10), the rotation matrix generator is written as bold, $\mathbf{R}(\cdot)$. However, it is more common to use a non-bold notation for functions. Therefore, it would be clearer to write it as $R(\cdot)$ without bold font.

**Limitations:**

I do not observe any major limitations in the proposed method.

**Strengths And Weaknesses:**

[Strengths]
1. The proposed method is lightweight and efficient, requiring only four parameters, which makes it potentially suitable for real-world and real-time applications. As shown in Figure 1 and Table 1, the proposed method achieves the best accuracy with significantly less computation compared to the baseline methods.

2. This paper contains a short, but gentle review of unit quaternions in Section 3.1. This is helpful for readers who may not be familiar with quaternions, as it provides the necessary background to understand the proposed method.

[Weaknesses]
Overall, the paper is well written and clearly presents the proposed method and experimental results. Only minor issues are noted below:

1. The tables present the performance of the proposed method together with baseline methods, and the proposed method is highlighted in bold. However, it is more common to highlight the best-performing results in bold. Therefore, the current convention may lead to misunderstandings when interpreting the performance tables. (Please refer to the Table 2.)

---

> ### Author Rebuttal · Authors · 2026-03-31
>
> **Weaknesses**
>
> > 　　We sincerely appreciate your positive feedback on the lightweight efficiency of QPoint, the clarify of the quaternion background introduction, as well as the overall presentation and experimental completeness of this paper. We are greatly encouraged by your recognition of the innovation, rigor and significance of our work.
> >
> > 　　Regarding the formatting convention for result tables, we appreciate your kind correction. Simply bolding only our own method in the tables could indeed cause confusion. In the revised version, we will follow the standard practice of bloding the best-performing results and underlining the second-best results across all tables to ensure clarify.
>
> **Questions 1**
>
> > 　　We greatly appreciate you raising this important observation that our performance on some segmentation tasks in Table 2 does not surpass several baseline methods.
> >
> > 　　First, as a standard benchmark for fine-grained part segmentation, ShapeNetPart dataset presents several challenging characteristics, including ambiguous part boundaries, a large number of small parts, and significant intra-class structural variations. These properties  demand much stronger local fine-grained feature extraction than classification tasks.
> >
> > 　　Second, part segmentation requires precise perception of tiny parts and narrow boundaries. Since part shapes of the same object vary greatly, dedicated local detail calibration modules are required, which rely on more parameters for feature extraction. Specialized methods **(e.g., PCM)** adopt substantially larger and more complex architectures with dedicated modules optimized for boundary details. In contrast, QPoint is designed with the core goals of extreme lightweightness and rotation robustness. Despite using only **21.6\%** of the **Parameters** and **9\%** of the **FLOPs** compared to PCM, QPoint has achieved an instance mIoU of **87.0\%**, approaching state-of-the-art baselines. This result fully demonstrates the optimal trade-off between efficiency and accuracy for lightweight models on fine-grained segmentation tasks.
> >
> > 　　We will further add a discussion of these results in the comparative experiments section for ShapeNetPart in the revised manuscript. In future work, we plan to incorporate lightweight multi-scale local perception and scene-level context modules to further improve segmentation performance while preserving rotation robustness and efficiency.
>
> **Questions 2**
>
> > 　　Thank you so much for your suggestion! "vot" is the abbreviation for "voting", referring to the voting strategy used during inference. In the revised version, we will replace it with the full term "voting" in the tables and add a clarify note to improve readability.
>
> **Questions 3**
>
> > 　　Thank you so much for this helpful formatting suggestion! In the revised manuscript,  We will update Equations (3) and (10) by using non-bold front for the matrix generation function to align with standard mathematical conventions.

---

> > ### Author Rebuttal · Reviewer_RgaX · 2026-04-01
> >
> > I appreciate the authors' effort to clarify my comments. All my concerns are completely resolved. I also realized that I did not checked the experimental results compared with Point Transformer v3 (PTv3), which was raised by other reviewers. I also think that the performance of PTv3 must be represented in Figure 1 together.

---

> > > ### Author Response · Authors · 2026-04-01
> > >
> > > We greatly appreciate the reviewer's valuable comments！
> > >
> > > Regarding the relevant experiments, we have followed the same approach as in our response to **Question 1** from **Reviewer xhWW**. We have conducted experiments on **PTv3** and updated the corresponding results in **Table 1** and **Table 2** in the revised manuscript.
> > >
> > > We sincerely thank you again for your high recognition of our work！

---

### Decision · Program_Chairs · 2026-04-30

**Decision:**

Accept (regular)

**Comment:**

The paper proposes a feature learning approach for 3D point clouds. After the initial review round, the paper received mixed ratings (4, 3, 2, 6) and was current on the fence. Reviewers required some clarifications regarding some of the methodological aspects of the work, and were concerned regarding missing experimental comparisons. The rebuttal solved most of the initial concerns, and the reviewers who have been more active in the discussion stage and that provided sound comments on the submission (xhWW, dn1B, RgaX) have afterwards landed on a consensus for acceptance. The AC agrees with their comments and recommends to accept this work - congratulations!